# Evaluating the Representation of Arctic Cirrus Solar Radiative Effects in the IFS with Airborne Measurements

Johannes Röttenbacher[1], André Ehrlich[1], Hanno Müller[1], Florian Ewald[2], Anna E. Luebke[1], Benjamin Kirbus[1], Robin J. Hogan[3, 4], and Manfred Wendisch[1]

[1]Leipzig Institute for Meteorology, Leipzig University, Leipzig, Germany
[2]Deutsches Zentrum für Luft- und Raumfahrt, Institut für Physik der Atmosphäre, Oberpfaffenhofen, Germany
[3]European Centre for Medium-Range Weather Forecasts, Reading, United Kingdom
[4]Department of Meteorology, University of Reading, Reading, United Kingdom

**Correspondence:** Johannes Röttenbacher (johannes.roettenbacher@uni-leipzig.de)

**Abstract.** In two case studies, airborne measurements of broadband solar irradiances above and below Arctic cirrus are compared to simulations of the European Centre for Medium-Range Weather Forecasts' (ECMWF's) Integrated Forecasting System (IFS) using offline runs of ECMWF's operational radiation scheme "ecRad". Furthermore, independent of the solar irradiances, cirrus properties are derived from collocated airborne active remote sensing observations to evaluate the optical and microphysical parameterizations in ecRad. The data were collected in the central Arctic over sea ice ($81°$ - $90°$ North) with instrumentation installed aboard of the High Altitude LOng range research aircraft (HALO) during a campaign in March and April 2022. Among others, the HALO instrumentation included upward and downward-looking pyranometers to measure broadband solar irradiances, a cloud radar, and a multi-wavelength water vapor differential absorption lidar. Extended horizontal flight legs above and below single-layer cirrus were performed. The solar radiation measurements are used to evaluate ecRad in two case studies of optically thin and thick cirrus with an average transmissivity of 0.9 and 0.6, respectively. Different ice optics parameterizations optionally available in ecRad are applied to test the match between simulation and measurements. Furthermore, the IFS-predicted ice water content and ice effective radius are replaced by values retrieved with the radar and lidar. The choice of ice optics parameterizations does not significantly improve the model-measurement agreement. However, introducing the retrieved ice microphysical properties brings measured and modelled irradiances in closer agreement for the optically thin cirrus, while the optically thick cirrus case is simulated as too thick. It is concluded that the ice water content simulated by the IFS is realistic and that the missmatch between observed and simulated solar irradiances mostly originates from the assumed or parameterized ice effective radius.

# 1 Introduction

Cirrus modifies the radiative energy budget of the Arctic atmosphere and surface (Hong and Liu, 2015; Marsing et al., 2023). Their total radiative effect is determined by the balance of a solar warming and a terrestrial cooling. In the terrestrial spectrum, cirrus warm the atmosphere below the cloud layer as they absorb most of the upward terrestrial radiation emitted by the surface and the atmosphere below the cloud, but usually emit at a lower temperature than the surface. This warming can be offset by the reflection of downward solar radiation at cloud top. The balance between reflected and transmitted solar radiation depends on the optical thickness of the cloud (Liou, 1986; Lynch, 2002). Ice crystal shape also plays an important role, especially for optically thin cirrus where single scattering processes dominate (Wendisch et al., 2005, 2007; Eichler et al., 2009; Baran, 2012).

In contrast to tropic and mid-latitude cirrus, the radiative effect of Arctic cirrus, which we define to occur north of the Arctic circle at $66°$ N, is strongly influenced by the bimodality of the surface albedo (open ocean vs. sea ice) and the usually low sun. During polar night with no solar radiation present in the Arctic cirrus warms the atmosphere and surface below the cloud (Hong and Liu, 2015). The magnitude of this warming depends on the surface temperature and its emissivity, which appear very heterogeneous depending on the sea ice cover and amount and size of leads and melt ponds varying as a function of season and latitude (Light et al., 2022). During polar day, the solar radiative effect of Arctic cirrus becomes more important. However, the solar zenith angle is still large leading to longer paths of the downward solar radiation through the cirrus. Furthermore, three-dimensional radiative effects such as upward and downward trapping and escape due to horizontal photon transport (Várnai and Davies, 1999) should be considered.

Because of the importance, Arctic cirrus should be represented realistically in numerical weather prediction (NWP) models. For this purpose, the atmospheric radiative transfer should be described appropriately in NWP models. However, especially the radiative properties of ice clouds are difficult to represent in such radiative transfer models due to the complexity of microphysical properties of ice crystals (Wolf et al., 2018; Lawson et al., 2019) and complex shape effects (Baran, 2012). These properties and shapes are commonly parameterized as a function of prognostic variables, such as temperature and ice water content (IWC) (Fu, 1996; Fu et al., 1998; Yi et al., 2013; Baran et al., 2014, 2016). For this purpose, the scattering and absorption properties of ice crystals of different shapes and sizes are calculated under certain assumptions (Yang et al., 2014). Together with particle size distributions (PSDs) bulk optical properties of clouds are then applied in the radiation scheme (Ebert and Curry, 1992).

The Integrated Forecasting System (IFS) operated by the European Centre for Medium-Range Weather Forecasts (ECMWF) uses the radiation scheme "ecRad" (Hogan and Bozzo, 2018). ecRad calculates the solar irradiances in 14 solar wavelength bands using a two-stream solver. The radiative transfer through cirrus is simulated utilizing the ice optics parameterization developed by Fu (1996) for the solar spectral bands. The parameterization for the solar radiative properties uses in situ measurements of PSDs from 28 flights in the mid-latitudes and tropics. No PSD data from Arctic cirrus is included. Assuming hexagonal ice crystals, Fu (1996) defines a generalized effective size ($D_{\mathrm{ge}}$) and IWC as the only two input parameters. Based on that, Fu (1996) parameterize the wavelength-dependent single scattering properties (extinction coefficient, single scattering

albedo, asymmetry parameter). They show that $D_{ge}$ relates to the total cross-sectional area of ice particles per unit volume, which makes the parameterization of the extinction coefficient and the single scattering albedo independent of particle shape. Thus, Fu (1996) conclude that the parameterization is suitable for the application in global climate models. However, it is also

mentioned that the asymmetry parameter is sensitive to the particle shape, which is not considered in this parameterization. Further studies showed that the surface roughness of ice crystals may also play an important role for their optical properties, especially for the scattering phase function and the asymmetry parameter (Eichler et al., 2009; Tang et al., 2017; Järvinen et al., 2018). The implementation of an optimized parameterization of the asymmetry parameter is, therefore, still an ongoing process in terms of the IFS's treatment of radiative transfer through cirrus.

The ice optics parameterization defined by Fu (1996) requires IWC and $D_{ge}$ as input. However, only the IWC is a prognostic variable in global models such as the IFS. Therefore, a parameterization of $D_{ge}$ is required. Fu (1996) showed that $D_{ge}$ can be related to the ice effective radius ($r_{eff}$) defined by Foot (1988) and Francis et al. (1994). This relation is applied by Sun and Rikus (1999) to parameterize $r_{eff}$ as a function of IWC and temperature using the PSD parameterization from McFarquhar and Heymsfield (1997). To evaluate the parameterization, Sun and Rikus (1999) used it together with the parameterization by

Fu (1996) to simulate the solar irradiances for two mid-latitudinal case studies. For these case studies in situ measurements of solar irradiances and $D_{ge}$ are available (Kinne et al., 1997). Comparing the simulations and observations yields agreement mostly within the measurement uncertainty. After comments by McFarquhar (2001), Sun (2001) revised the parameterization to account for a singularity. The resulting parameterization of $r_{eff}$, together with the ice optics parameterization by Fu (1996) has been used within the IFS since 2007 (Hogan and Bozzo, 2018).

The prospects for ice optics parameterizations increased in recent years due to the availability of increasingly complex scattering models (Yang et al., 2014) and more powerful computers. Furthermore, more in situ measurements of ice crystal particles in cirrus at different altitudes, latitudes and temperatures became available (Baran, 2012; Luebke et al., 2013; Krämer et al., 2020). However, only few in situ data of ice crystals in the Arctic have been reported. De La Torre Castro et al. (2023) analyzed in situ measurements of ice particles from the Cirrus in High Latitudes (CIRRUS-HL) aircraft campaign, which took place in

June and July 2021. A limited number of flights ($7.8\,\mathrm{hours}$ within cirrus north of $60°$ North) provided valuable data sets of Arctic cirrus. The authors compared mid- and high-latitude cirrus with respect to the ice crystal number concentration, mean effective diameter and IWC. They found that cirrus formed in the high-latitudes have a lower ice crystal number concentration and a larger ice effective diameter, compared to cirrus formed in the mid-latitudes, which are commonly used to derive the ice optics parameterizations cited above.

It is therefore important to test the capabilities of the current IFS simulations to realistically represent the influence of cirrus on solar radiative transfer in the Arctic. One approach to evaluate the performance of the IFS, and ecRad in particular, has been shown by Wolf et al. (2020) and Müller et al. (2024), who used airborne observations of solar irradiance and remote sensing data to evaluate the representation of ice-topped clouds over the North Atlantic and Arctic low-level clouds, respectively. Here, we build on this method to quantify the performance of ecRad concerning Arctic cirrus. To separate the possible sources

of uncertainties in the simulations, we utilize a microphysical retrieval of cirrus properties to perform sensitivity studies with respect to the predicted IWC by the IFS. The aircraft campaign, instruments and measurements used in this study are introduced

in Sect. 2. In Sect. 3, the setup of the radiative transfer simulations and the evaluation strategy is explained. The results of the evaluation are presented in Sect. 4 and 5 followed by a summary and conclusions in Sect. 6.

## 2 Airborne observations of Arctic cirrus

### 2.1 Measurements

Evaluating the performance of ecRad with respect to Arctic cirrus requires direct measurements of the radiative budget above and below cloud. In the remote central Arctic, such measurements were collected by the High Altitude LOng range research aircraft (HALO) during the HALO–$(\mathcal{AC})^3$ (ArctiC Amplification: Climate Relevant Atmospheric and SurfaCe Processes, and Feedback Mechanisms) campaign (Wendisch et al., 2024). HALO was stationed in Kiruna, Sweden and conducted 18 research flights between March and April 2022 into the Fram Straight and over the central Arctic Ocean. HALO was instrumented with its cloud observatory payload (Stevens et al., 2019) consisting of, among others, the WAter vapor Lidar Experiment in Space (WALES, Wirth et al., 2009) and the HALO Microwave Package radiometer system (HAMP, Mech et al., 2014), including a 35 GHz cloud radar. For measurements of atmospheric temperature and humidity profiles, numerous dropsondes were released. To measure the upward and downward broadband irradiances, HALO was equipped with the Broadband AirCrAft RaDiometer Instrumentation (BACARDI, Ehrlich et al., 2023), which consists of two sets of Kipp & Zonen CMP22 pyranometers and CGR4 pyrgeometers. They measure irradiances in the solar $(0.3 - 3\,\mu m)$ and terrestrial $(3 - 100\,\mu m)$ wavelength range. For this study, the solar irradiance measurements were analyzed.

The data processing of the pyranometer including corrections for the sensor sensitivity, thermal dynamic offsets, sensor inertia and aircraft attitude are described by Ehrlich et al. (2023), who also discuss the measurement uncertainties of the pyranometers. For operation in high latitudes with low ambient temperature conditions such as during HALO–$(\mathcal{AC})^3$, the temperature dependency of the sensor sensitivity has to be considered. During HALO–$(\mathcal{AC})^3$ ambient temperatures around $-55\,°C$ were common at flight altitude, exceeding the calibration certificate by the manufacturer. Therefore, the uncertainty of the measured irradiances is higher than reported by Ehrlich et al. (2023).

Additionally, the large solar zenith angles (low sun) pose another challenge in high latitudes. Uncertainties due to the imperfect angular response of the pyranometers reach up to $1\,\%$ for solar zenith angles larger than $80°$ (Ehrlich et al., 2023) and the correction of the aircraft attitude introduces additional uncertainties (Wendisch et al., 2001). Similar to Ehrlich et al. (2023), the performance of BACARDI during HALO–$(\mathcal{AC})^3$ was evaluated against cloud-free radiative transfer simulations. For the solar downward irradiance measured at altitudes of $10\,km$ and higher, the uncertainties are estimated with about $\pm 5\,\%$.

### 2.2 Two case studies

For the interpretation of the radiative properties of cirrus, a cloud-free atmosphere below the cirrus is advantageous because the effect of the cirrus and low clouds are hard to distinguish otherwise. During HALO–$(\mathcal{AC})^3$ research flight (RF) 17 on 11 April 2022 and RF 18 on 12 April 2022 fulfilled these conditions of a single layer cirrus. The tracks of the two flights together

with the respective high cloud cover and the sea ice edge as predicted by the IFS $0\,\mathrm{UTC}$ run from 11 and 12 April 2022 are shown in Fig. 1 (a) and (b). The flight sections of the analyzed cases are highlighted in orange in the far north of the flight track. Along each of those flight legs remote sensing measurements were performed above and below cirrus. In addition to the IFS forecasts Fig. 1 (c) and (d) show the false color corrected reflectance product from the Moderate Resolution Imaging Spectroradiometer (MODIS) on the Terra satellite using Band 3, 6 and 7. This band combination is sensitive to ice and snow and allows to distinguish cirrus, visible as white to slightly orange filaments, from sea ice, which appears in dark orange.

Both cases happened under similar synoptic conditions. On 09 April 2022, a low pressure system over Scandinavia advected warm and moist air into the Fram Strait. Due to a convergence with a cold air mass, that had its origin in Greenland, a vertical wind shear developed with low-level northeasterly winds and southeasterly winds at levels above $700\,\mathrm{hPa}$. This southeasterly flow transported the initial moisture of the observed cirrus into the central Arctic. Because of a cold air outflow from the Greenland ice sheet into the central Arctic, the northbound air mass had to slowly ascend and formed the isolated cirrus with no low level clouds below. The backward trajectories ending in the observed location and altitude of the cirrus shown in Fig. 1 were calculated with the Lagrangian analysis tool (LAGRANTO, Sprenger and Wernli, 2015) based on data from the fifth version of the ECMWF's atmospheric reanalysis (ERA5, Hersbach et al., 2020). The trajectories indicate that the majority of the air mass featuring the cirrus was transported from the south into the central Arctic. Some of the air mass featuring the cirrus on 11 April 2022 also originated in the eastern central Arctic. On 12 April 2022 the whole sampled air mass came with the poleward moisture transport through the Fram Strait. The IFS high cloud cover (shown as blue shading in Fig. 1) indicates that the observed cirrus was part of a larger cirrus field. This large cirrus field can also be seen in the satellite product depicted in Fig. 1 (c) and (d), which is a combination of overflights from the Terra satellite between $14\,\mathrm{UTC}$ and $20\,\mathrm{UTC}$ on the respective case study date. In Fig. 1 (c), depicting the situation during RF 17, the edge of the cirrus field can be seen close to the radiosonde dropped at $10:42\,\mathrm{UTC}$. Here the cirrus is optically thin while further west on the flight track the optical thickness increases. As indicated by the satellite image, the observations took place at the edge of the cirrus field, which is stretching southwards east of the Greenland coast similar to the IFS forecast. This cirrus field persisted and can be seen again on the 12 April 2022 in Fig. 1 (d). Down to $86\,^\circ\mathrm{N}$ the cirrus field is rather compact and part of the same air mass. Bigger sections of very optically thick cirrus only appear further south and reach all the way to the sea ice edge close to the radiosonde dropped at $09:39\,\mathrm{UTC}$.

Due to the large scale lifting of the air mass, the cirrus is rather homogeneous on a large scale. In combination with the uniform shape of the trajectories, this suggests that the observations from the case study areas are representative for this cirrus field and cirrus properties can be assumed to be stable during the one hour flight patterns. This justifies the combination of observations from the above-cloud and below-cloud flight legs, which are separated by about 15 minutes. On a smaller scale, however, horizontal heterogeneities were present.

## 2.3 Cirrus radiative properties

Figure 2 shows solar broadband irradiance measurements of the above and below-cloud section from RF 17 and RF 18 along the flight track. Data has been filtered for aircraft attitude changes during turns, descents, and ascents. The transmissivity of

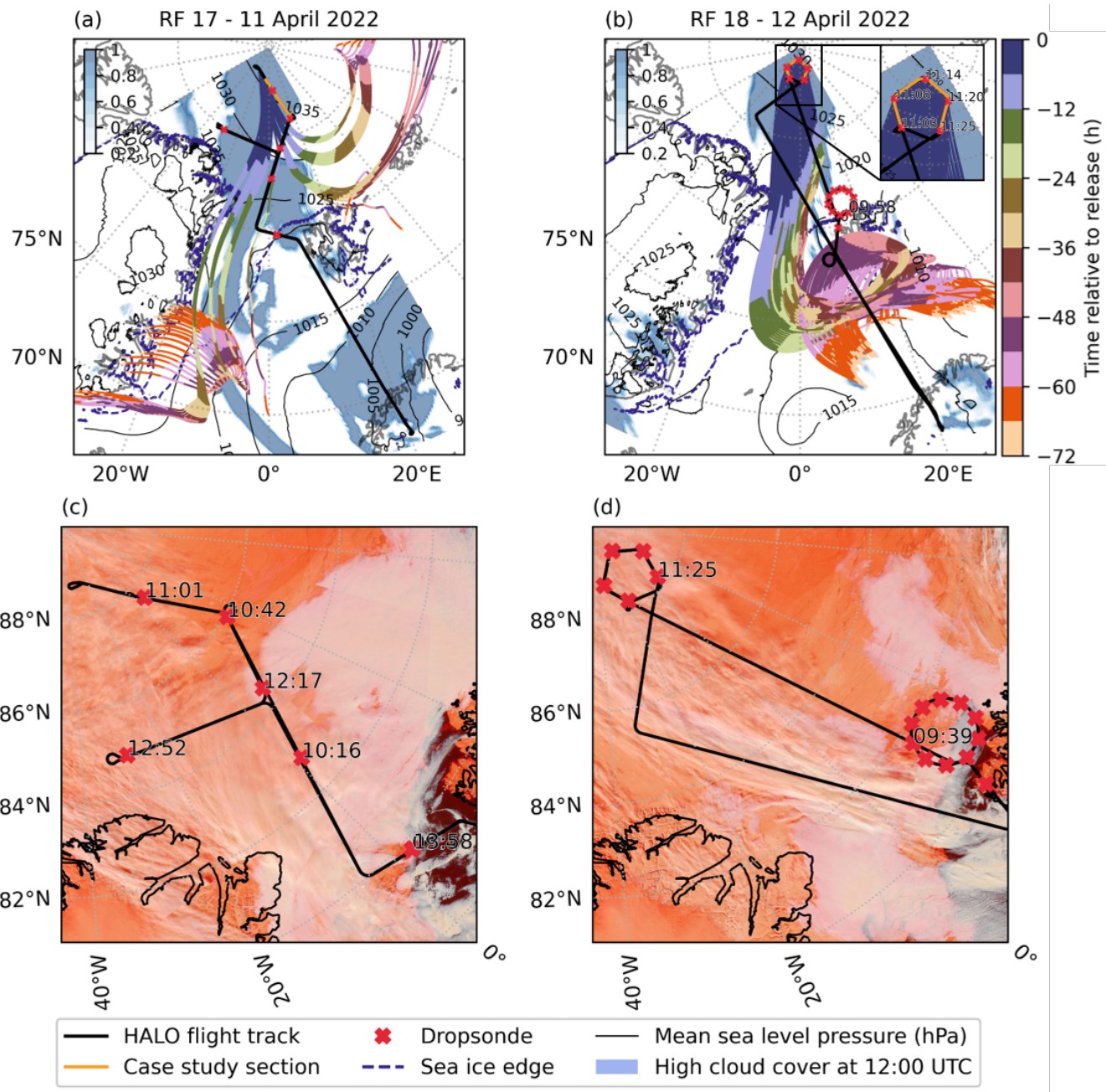

**Figure 1.** Map of flight tracks with IFS predicted high cloud cover for 12 UTC, sea ice edge (80 % sea ice cover), mean sea level pressure isolines, dropsonde locations (red crosses), highlighted case study regions (orange), and LAGRANTO backward trajectories for (a) RF 17 and (b) RF 18. The box in panel (b) shows a zoom of the case study region with the above and below-cloud flight sections for RF 18. (c) and (d) False color corrected reflectance from MODIS on Terra using Band 3, 6 and 7 for RF 17 and RF 18, respectively, as provided by the Global Imagery Browse Services (GIBS) from NASA.

the cloud is calculated as the ratio between the below-cloud measurements and the cloud-free downward irradiance at around $11\,\mathrm{km}$, provided by the ecRad simulations described in Sect. 3. To facilitate understanding, the x-axis shows the distance traveled by HALO from the start to the end of the above-cloud section. The two flights show distinctively different structures in the measurements owing to the different flight patterns. In Fig. 2 (a) the straight flight leg towards the northwest and the corresponding increase in the solar zenith angle from $77.6°$ to $79.8°$ is reflected in the decrease in solar downward irradiance from $250\,\mathrm{W\,m^{-2}}$ to $225\,\mathrm{W\,m^{-2}}$. During the pentagon flight pattern in RF 18 the solar zenith angle was generally larger than $79°$ and thus changed less than during RF 17. This results in a lower average solar downward irradiance of $206\,\mathrm{W\,m^{-2}}$ for RF 18 compared to $238\,\mathrm{W\,m^{-2}}$ for RF 17. The below-cloud measurements reveal high variability in the transmissivity which is linked to the optical depth of the cirrus. In RF 17 the downward solar irradiance varied with a standard deviation of $16\,\mathrm{W\,m^{-2}}$ corresponding to a change of transmissivity between $0.69 - 0.93$. The cirrus in RF 18 showed slightly less variability ($11\,\mathrm{W\,m^{-2}}$ standard deviation) and was less transmissive ($0.49 - 0.71$).

## 3 Evaluation strategy for the IFS and ecRad

### 3.1 IFS and ecRad setup

The airborne observations are used to evaluate the IFS of the ECMWF. The solar broadband irradiances are compared with the simulations of the radiation scheme ecRad. The IFS forecast is fed into version 1.5.0 of ecRad, which is run in an offline mode to allow for sensitivity studies by exchanging specific IFS forecast variables and vary the applied ice optics parameterization. This approach is adapted from Wolf et al. (2020) and Müller et al. (2024). Instead of the operational Monte Carlo integration of the Independent Column Approximation (McICA, Pincus et al., 2003), the Speedy Algorithm for Radiative Transfer through Cloud Sides (SPARTACUS, Hogan et al., 2016; Schäfer et al., 2016) is used as a solver because it provides the spectral irradiances at all model levels and has tools available to parameterize 3D radiative effects including the radiative transfer through cloud sides and entrapment (Hogan et al., 2019). The spectral resolution of the ecRad simulations depends on the chosen gas absorption model, which in this study is the Rapid Radiative Transfer Model for General Circulation Models (RRTM-G, Mlawer et al., 1997). Thus, the irradiance is calculated for 14 solar bands listed in Table 1.

We employ the IFS data from the operational octahedral reduced Gaussian grid (O1280) based on the IFS cycle 47R1 with 137 vertical levels. The hourly IFS forecast initialized at $0\,\mathrm{UTC}$ of the corresponding flight day is used, and all relevant input variables for ecRad are extracted along the flight path of HALO. One problem, that needs to be accounted for, is the scale mismatch between the measurements and the model output. The BACARDI irradiance measurements are recorded at $10\,\mathrm{Hz}$ time resolution resulting in a measurement every $20\,\mathrm{m}$ assuming an average aircraft speed of $200\,\mathrm{m\,s^{-1}}$ for HALO. Although BACARDI samples the whole hemisphere, $95\,\%$ of the signal stem from a footprint of about $500\,\mathrm{km^2}$ at $3\,\mathrm{km}$ distance from the target, which is the approximate flight altitude above cirrus for the case studies. The horizontal resolution of the IFS in the Arctic varies slightly between $8\,\mathrm{km}$ and $9.5\,\mathrm{km}$. Therefore, the measurements are averaged to one minute resolution to match the spatial resolution of one IFS grid cell. For the same reason, simulations are performed every minute along the flight track.

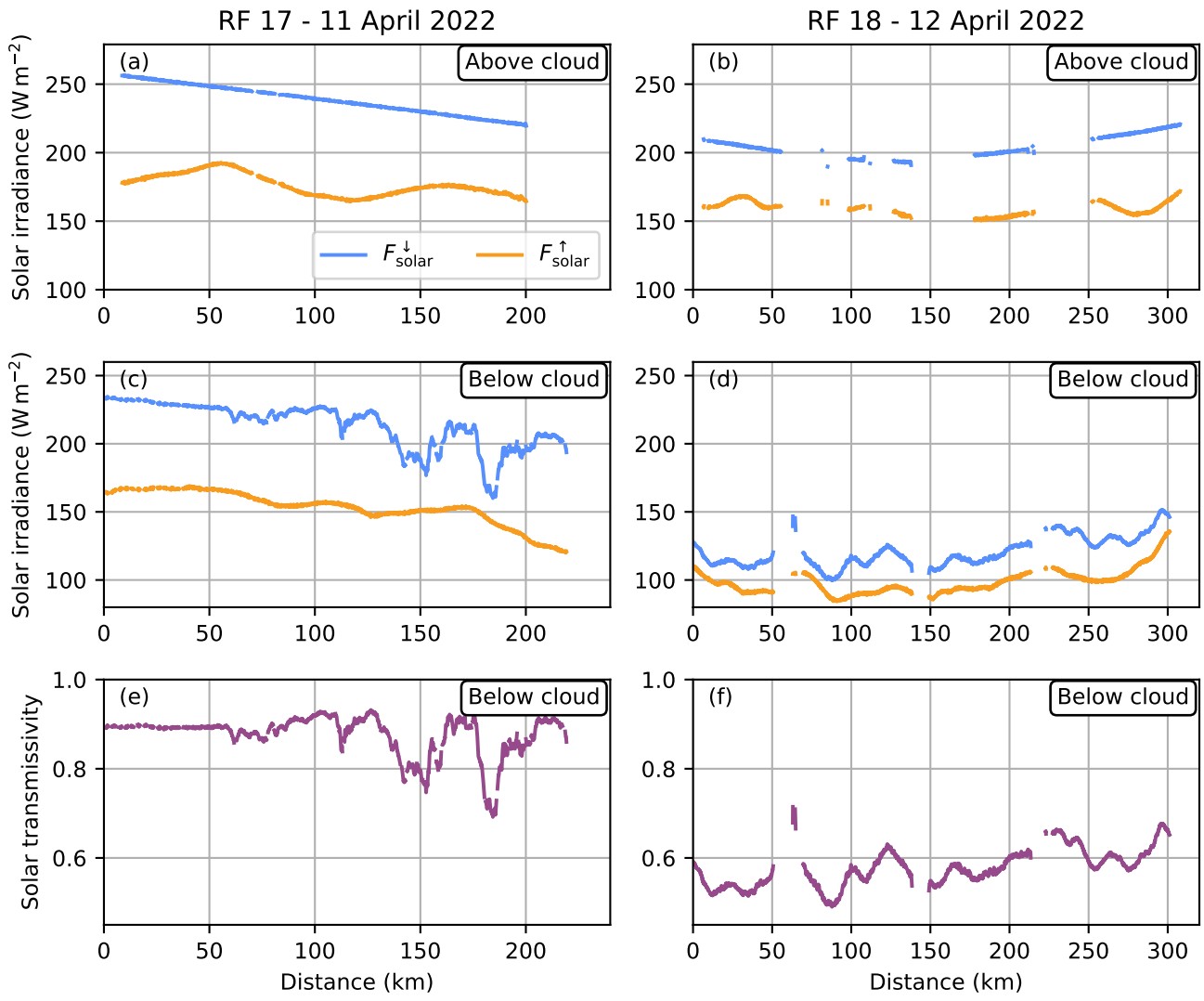

**Figure 2.** Measured downward and upward solar irradiance from BACARDI for the (a, b) above and (c, d) below-cloud sections of (a, c) RF 17 and (b, d) RF 18. Panels (e) and (f) show the solar transmissivity below cloud. The x-axis shows the distance traveled by HALO from the start to the end of the above-cloud section.

Regarding the time mismatch, the IFS data is not averaged between time steps as this would assume a linear change for all model parameters, which might be true for some but certainly not for all quantities such as relative humidity and IWC. This results in a maximum offset of $30\,\mathrm{minutes}$ between any measurement and the next closest forecast time step. For the case studies presented here this means that the above-cloud sections are covered by the $11\,\mathrm{UTC}$ time step and the below-cloud sections are covered by the $12\,\mathrm{UTC}$ time step, with the switch happening either during the descent to the below-cloud section or just afterwards. The solar zenith angle is set to the time and location of HALO for each ecRad simulation step. To account for a possible slight offset of the location of the clouds between model and measurement, the ten closest grid points to the flight path are selected, covering a circular area between $616\,\mathrm{km}^2$ and $1050\,\mathrm{km}^2$. The same solar zenith angle is used for the simulation of the ten closest grid points.

For the trace gases CFC-11, CFC-12, $CO_2$, $CH_4$, $N_2O$ and $O_3$ a monthly mean climatology is used depending on latitude and pressure level. The climatology is based on the reanalysis produced by the Monitoring Atmospheric Composition and Climate project (MACC, Inness et al., 2013) for $CO_2$ and $CH_4$, the Cariolle model (Cariolle and Déqué, 1986; Cariolle and Teyssèdre, 2007) for CFC-11, CFC-12 and $N_2O$, and the Copernicus Atmospheric Monitoring Service (CAMS) Interim reanalysis (Flemming et al., 2017) for $O_3$. The incoming solar irradiance at top of atmosphere is also prescribed in the simulations using a value of $1360.8\,\mathrm{W\,m^{-2}}$ provided by Kopp and Lean (2011) and is adjusted for the Earth-Sun-distance at noon of each flight day. To keep the focus on the cloud properties, aerosol particles are turned off in the simulations. A detailed Fortran namelist with all options and an example input file is available in the GitHub repository accompanying this paper (https://github.com/radiation-lim/roettenbacher_etal_2024, last accessed 17.05.2024).

### 3.2 Surface albedo parameterization

In the operational mode, when ecRad is run within IFS, the IFS host system provides the surface albedo. In the offline mode, that is used here, this parameter has to be provided. Therefore, following the operational setup, the surface albedo along the flight track is calculated by combining an open ocean albedo with the sea ice albedo parameterization by Ebert and Curry (1993) accounting for the present ratio of direct and diffuse irradiance. The two albedo values are thereby weighted according to the IFS predicted sea ice fraction. A further differentiation of the ocean albedo is made by using the solar zenith angle dependent parameterization from Taylor et al. (1996) for the direct solar surface albedo, while the diffuse solar surface albedo is fixed to a value of $0.06$. The ocean albedo is assumed to be constant with wavelength, while the sea ice albedo parameterization provides a monthly mean value for six solar bands. Linear interpolation is performed in time, treating each of the twelve monthly means as the instantaneous value at the 15th day of each month. These interpolated values are then internally mapped by ecRad to the 14 solar bands defined by the RRTMG using a weighted average according to the overlap of the six albedo bands with the RRTMG bands. For the winter months, the IFS assumes the surface albedo for dry snow and for the summer months the albedo of bare ice according to Ebert and Curry (1993). Although the case studies appeared in April we still follow the assumptions for the winter months as the central Arctic wasn't affected by thawing during the case studies. Thus, the assumption of bare ice is not yet justified. The resulting mean solar surface albedo for each RRTMG solar band for the case study period of RF 17 and RF 18 are given in Table 1.

**Table 1.** ecRad solar bands as defined by the RRTM-G and the corresponding mean solar surface albedo for the case study periods of RF 17 and RF 18.

| Band number | Wavelength ($\mu$m) | Mean solar surface albedo | |
| --- | --- | --- | --- |
| | | RF 17 | RF 18 |
| 1 | 3.08 - 3.85 | 0.025 | 0.025 |
| 2 | 2.50 - 3.08 | 0.025 | 0.025 |
| 3 | 2.15 - 2.50 | 0.180 | 0.180 |
| 4 | 1.94 - 2.15 | 0.249 | 0.250 |
| 5 | 1.63 - 1.94 | 0.249 | 0.250 |
| 6 | 1.30 - 1.63 | 0.249 | 0.250 |
| 7 | 1.24 - 1.30 | 0.249 | 0.250 |
| 8 | 0.78 - 1.24 | 0.789 | 0.791 |
| 9 | 0.63 - 0.78 | 0.898 | 0.901 |
| 10 | 0.44 - 0.63 | 0.972 | 0.975 |
| 11 | 0.34 - 0.44 | 0.972 | 0.975 |
| 12 | 0.26 - 0.34 | 0.972 | 0.975 |
| 13 | 0.20 - 0.26 | 0.972 | 0.975 |
| 14 | 3.85 - 12.20 | 0.025 | 0.025 |

### 3.3 Ice optics parameterizations

The choice of ice optics parameterization is crucial to simulate the Earth's radiative energy budget (Fu, 2007; Baran, 2012; Yi, 2022). With ecRad it is possible to switch between different parameterizations including the operational one from Fu (1996) (referred to as Fu-IFS), the one based on Yi et al. (2013) (referred to as Yi2013), and the most recent one by Baran et al. (2016) (referred to as Baran2016). The relevant details of the three parameterizations are summarized in Table 2. In general, all three parameterizations derive the extinction coefficient, single-scattering albedo and asymmetry parameter of a grid cell as a function of IWC and temperature. However, Fu-IFS and Yi2013 do not use temperature directly but instead need $r_{\text{eff}}$ as an explicit input. Thus, $r_{\text{eff}}$ is calculated in the operational mode of the IFS using the parameterization from Sun (2001). To account for the observation that ice crystals are generally larger in the tropics than in the mid-latitudes (Field et al., 2007), the minimum $r_{\text{eff}}$ predicted by the Sun (2001) parameterization is scaled with the cosine of the latitude. Due to the lack of in situ observations in the Arctic, this parameterization with latitude is extrapolated for high latitudes.

Baran2016, on the other hand, parameterize the bulk optical properties directly as a function of IWC and in-cloud temperature. Another difference to Fu-IFS and Yi2013 is the use of synthetic PSDs derived from the parameterization by Field et al. (2007). By that, the temperature range of the parameterization is extended down to $-80\,°\text{C}$, and the diagnostic $r_{\text{eff}}$ is not needed in the parameterization. Apart from the difference in input variables, the parameterizations are also based on different sets of in situ cirrus observations and assume different ice habits and mixtures (see Table 2).

**Table 2.** Main differences between the three available ice optics parameterizations in ecRad.

| Name | Particle size distributions | Ice crystal habit assumption | Input variables |
|---|---|---|---|
| Fu-IFS | 28 from mid-latitudes and tropics | randomly oriented hexagonal ice crystals | IWC and ice crystal effective radius |
| Yi2013 | 14 408 from 11 field campaigns (Heymsfield et al., 2013) | nine habits with a rough surface from a general habit mixture dependent on maximum diameter (droxtals, solid and hollow bullet rosettes, solid and hollow columns, plates, aggregate of solid columns, small and large aggregate of plates) (Baum et al., 2011) | IWC and ice crystal effective radius |
| Baran2016 | Synthetic PSDs using 20 662 combinations of IWC and in-cloud temperature using the parameterization developed by (Field et al., 2007) based on 14 000 PSDs | ensemble model of six ice crystal habits dependent on maximum dimension (hexagonal ice column, 6-branched bullet rosette, 3-, 5-, 8- and 10- monomer ice aggregates (Baran and Labonnote, 2007) | IWC and temperature |

## 3.4 VarCloud retrieval

To independently validate the prognostic IWC and the diagnostic $r_{\mathrm{eff}}$ used in the ice optics parameterizations, active remote sensing measurements from HALO are applied. Using synergistic radar and lidar measurements from HALO, the IWC and $r_{\mathrm{eff}}$ are retrieved using a technique developed by Ewald et al. (2021). The retrieval is based on a variational optimal estimation algorithm (VarCloud, Delanoë and Hogan, 2008). The algorithm iterates forward simulations of radar and lidar reflectivity until convergence between the simulated and measured signals is reached. The first guess of the cloud properties is provided using the radar-lidar mask with climatological profiles of number concentration and lidar ratio which are a function of temperature (Cazenave et al., 2019). The lidar backscatter is then calculated using the forward model developed by Hogan (2008) and look up tables of T-matrix (Mishchenko et al., 1996) calculations of soft spheroids for the radar backscatter. The forward simulations assume a constant relationship between the ice crystal mass $M$ and the volume of a sphere that encloses the maximum diameter $D_{\mathrm{max}}$ of the ice particle following the most recent update of the "Composite" $M$-$D_{\mathrm{max}}$ relationship by Cazenave et al. (2019). The particle shape is modeled as horizontally aligned oblate spheroids following Hogan et al. (2012). Following the approach by Delanoë et al. (2005), a normalized PSD is used. By integrating the visible extinction cross-section, which is approximated by two times the geometric cross-section, and the radar cross-section over this PSD, the visible extinction and radar reflectivity are obtained. IWC is retrieved in the same way using the ice crystal mass. From these values, $r_{\mathrm{eff}}$ is calculated using the relationship between IWC and visible extinction coefficient derived by Foot (1988). The product provides profiles of $r_{\mathrm{eff}}$ and

IWC of cirrus along the flight track at a $1\,\mathrm{second}$ time resolution (~200 m horizontal) and $30\,\mathrm{m}$ vertical resolution when the aircraft was above cloud.

## 4  Cirrus representation in the IFS

Before comparing solar broadband irradiances, the general representation of the synoptic condition and the cirrus in the IFS numerical weather forecast are evaluated by dropsonde and remote sensing observations from HALO. One cause of uncertainty in the representation of cirrus can be an incorrect prediction of the atmospheric state in the IFS. Fig. 3 shows the atmospheric temperature and humidity profiles from the IFS forecast for the case study period (above and below-cloud section) and the ones measured by the dropsondes released during the above-cloud sections of the two flights. It has to be mentioned that the IFS did assimilate the dropsondes of the previous flights. While no flight (no dropsondes) was performed the day before RF 17, the dropsondes released during RF 17 might have affected the forecast for RF 18.

The deviations between the IFS and the dropsonde temperature profiles are minor and only reach up to $6\,\mathrm{K}$ close to the boundary-layer inversion, where a slight mismatch of the inversion altitude is found. As can be seen from the temperature inversions at around $8\,\mathrm{km}$, HALO was well above the tropopause during both flights. The relative humidity ($\mathrm{RH_{ice}}$) profiles indicate a high vertical variability both in the IFS and in the observations. While the height of the humidity drop at the tropopause matches, the humidity decrease at cloud base is more variable and slightly differs from the observations. This strong decrease just at the tropopause means that the cirrus is capped by it. For RF 17, the IFS humidity profiles indicate a trend of decreasing cloud base along the flight track by showing higher values at lower altitudes. This is due to the cirrus thickening towards the end of the above-cloud section that is not fully captured by the dropsonde measurements. For RF 18, the layer of high humidity is more stable at cloud base (smaller flight area) but slightly higher than in the dropsonde observations. However, the model does not cover the full range of the measured $\mathrm{RH_{ice}}$, especially below cloud. A very prominent feature during RF 18 is that the model does not predict supersaturation with respect to ice inside the cirrus, which the dropsonde observes. This is due to the ice cloud scheme of the IFS, which allows for supersaturation with respect to ice in the clear sky portion of the grid cell but immediately converts supersaturation in the cloudy part into cloud ice thus limiting $\mathrm{RH_{ice}}$ to $100\,\%$. RF 17 does not feature many fully cloudy grid cells, and thus does not show this feature as prominent. The dropsondes show only a thin layer with $\mathrm{RH_{ice}}$ exceeding $100\,\%$. This indicates that a vertically thin cirrus was present during RF 17.

A more direct comparison of the cloud top and base altitude is achieved using radar and lidar measurements. Figure 4 shows the VarCloud lidar-radar cloud mask from HALO, the aircraft altitude, and the predicted cloud fraction of the IFS along the flight track of HALO. It can be seen that the IFS prediction mostly matches the actual location of the clouds in their vertical position. For RF 17, Fig. 4 (a), a slight overestimation of cloudiness can be seen at the beginning of the above-cloud section. Towards the end of the above-cloud section, the cloud becomes geometrically thicker earlier in the IFS but does not reach the same vertical extent as in the observations. Thus, the below-cloud section started inside the lower part of the cirrus. However, these cloud layers were not visible by eye and can only be inferred from the radar and lidar signals. For the comparison and the simulations this part of the below-cloud section is excluded. Figure 4 (b) shows that the cloud in RF 18 extends to

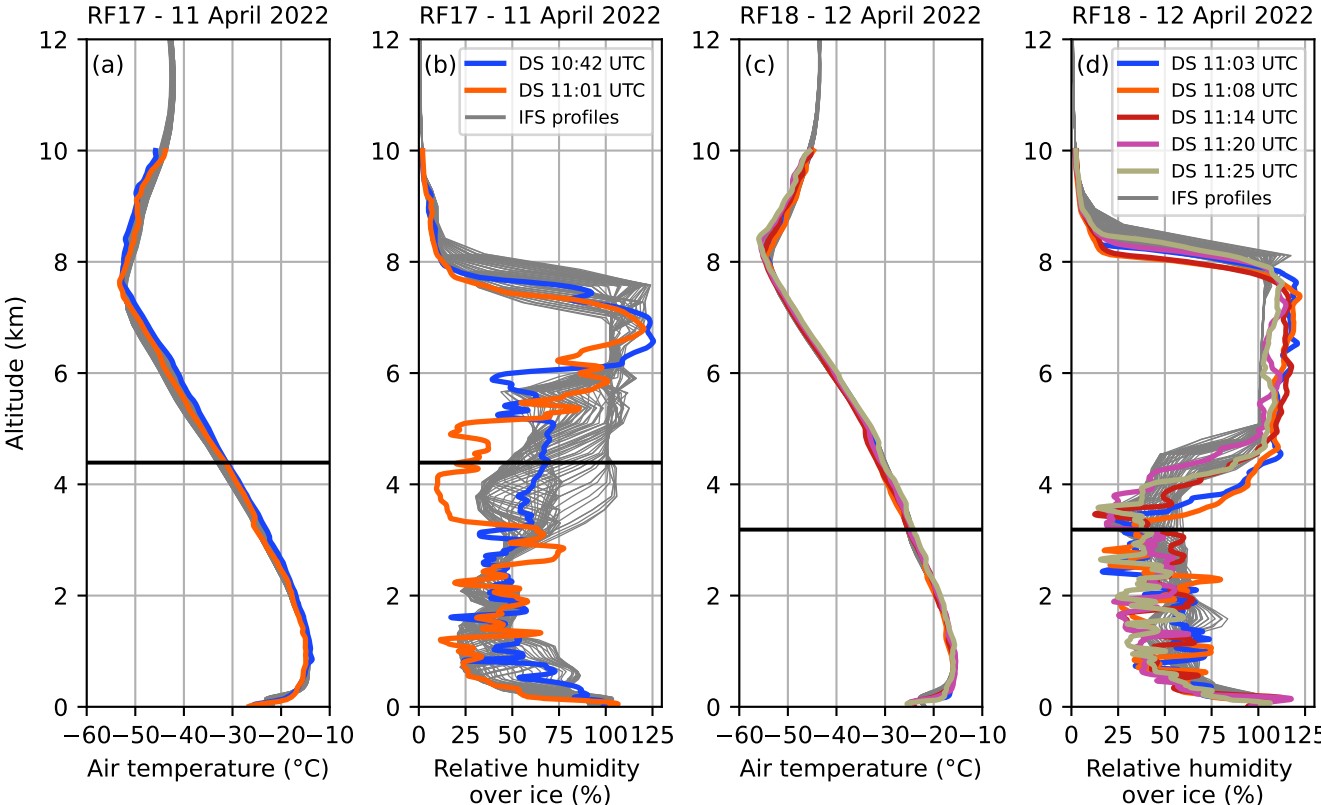

**Figure 3.** Atmospheric profiles of (a, c) air temperature and (b, d) relative humidity over ice from the IFS (grey lines) for the whole case study period (above and below-cloud section) along the flight track and the dropsondes (DS) deployed by HALO during the above-cloud section of (a, b) RF 17 and (c, d) RF 18. The black line indicates the flight altitude of HALO during the below-cloud section.

lower altitudes than predicted during the whole above-cloud section. It should be mentioned that the IFS does predict small
snow water content values (precipitating ice) well below the radar and lidar mask - for RF 18 even down to the surface -
yet the important variable for ecRad is the cloud fraction. If no cloud fraction is predicted in a grid layer no cloud optical
properties are computed. According to the IFS both flights also show a small amount of cloud particles close to the ground at
the beginning of the above-cloud sections. The cloud fraction for these cloud layers only show values below $0.15$, indicating
thin clouds linked to the surface temperature inversion. However, these clouds were not captured by the radar or the lidar, nor
by visual observations from the aircraft. Therefore, these cloud layers are removed from the IFS output for further analysis and
the upcoming simulations.

The dashed vertical line in Fig. 4 (a) and (b) indicates the time when the IFS forecast switches from the $11\,\mathrm{UTC}$ to the
$12\,\mathrm{UTC}$ time step. Thus, the change in cloud fraction shown here is mostly due to HALO flying through different grid cells.
To quantify, how strong the cirrus field changed between the two time steps of the IFS, Fig. 5 shows the distribution of the IFS
IWC at $11\,\mathrm{UTC}$ and at $12\,\mathrm{UTC}$ in the case study areas. RF 17 shows a shift towards smaller values from $11\,\mathrm{UTC}$ to $12\,\mathrm{UTC}$,

285

290

295

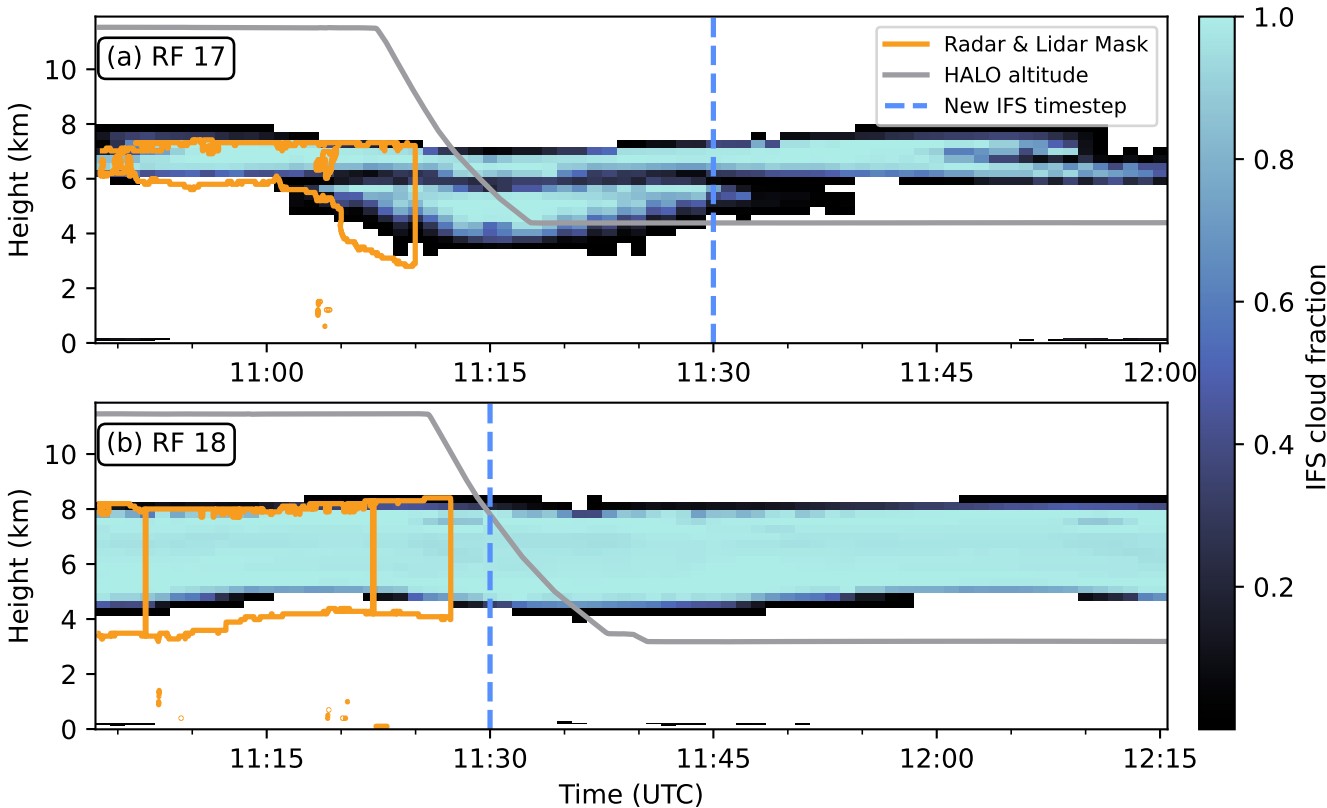

**Figure 4.** Comparison of lidar-radar cloud mask and IFS predicted cloud fraction for the case studies of (a) RF 17 and (b) RF 18.

meaning that the cloud is dissolving or moving out of the case study area according to the IFS. For RF 18 no noteworthy differences can be seen between the 11 UTC and 12 UTC time step.

## 5 Comparisons of measured and simulated solar downward irradiance

Flight sections of cloud-free measurements above the aircraft can be used to validate the downward irradiance measurements. Due to the lack of scattering and absorption by the remaining atmosphere above HALO, the radiative transfer simulations can be assumed to be precise. Remaining differences between simulation and measurement are assumed to stem from the instrument performance. Selected sections above 11 km of both RFs are compared and shown in Fig. 6. The validation shows a slight positive bias in the measurements and a small root mean square error of $3 - 5\,\mathrm{W\,m^{-2}}$, which is within the uncertainty range of BACARDI.

300

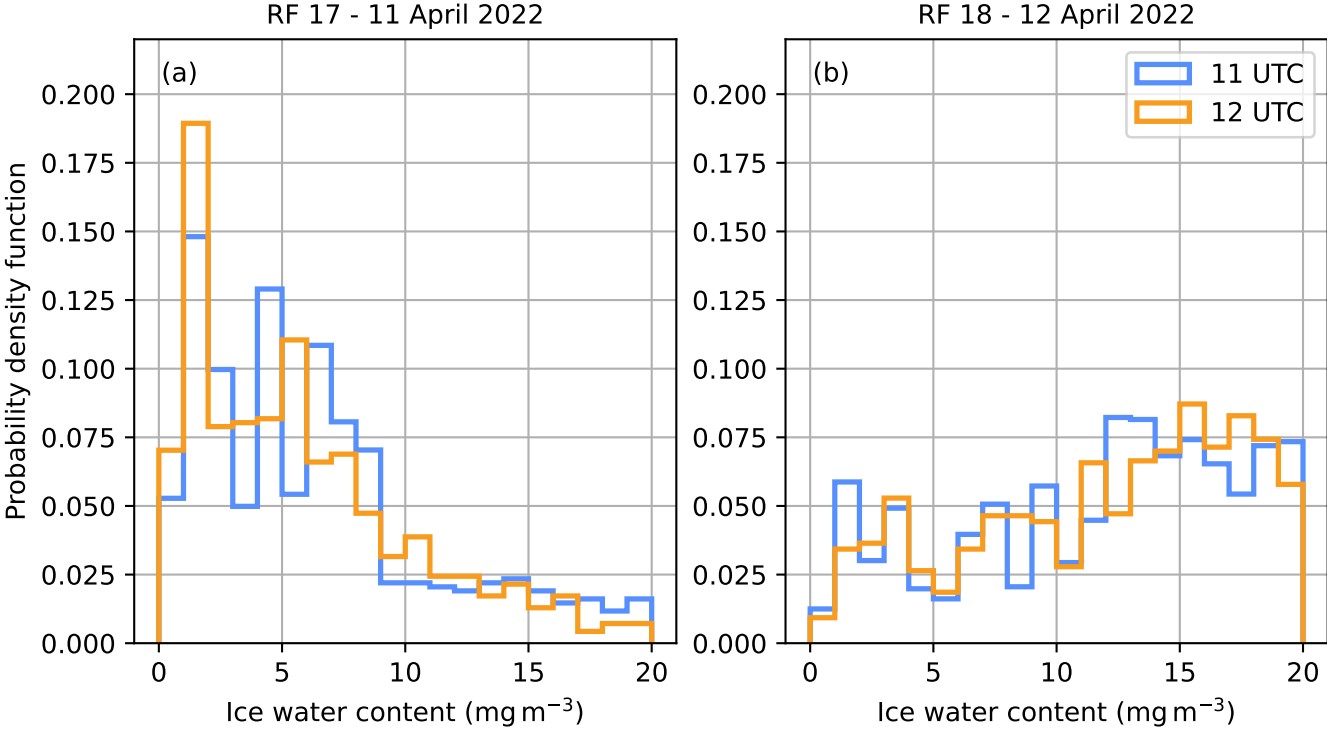

**Figure 5.** Probability density functions of the IWC for the 11 UTC and 12 UTC IFS time step for the IFS grid points with cloud fraction greater 0 in the case study areas of (a) RF 17 and (b) RF 18.

### 5.1 Sea ice albedo influence

A crucial parameter determining the radiative transfer in the Arctic is the sea ice. Our case studies were situated in latitudes close to $80°$ N over closed sea ice with only small ridges, leads and refrozen leads visible from the aircraft. These relatively small inhomogeneities of the sea ice are only represented in the sea ice fraction of IFS for RF 17, where it is between 0.989 and 0.999 in the area of the case studies. For RF 18 it is constantly 1. Although, the inhomogeneities are not imprinted in the time series of irradiance as they smooth out due to the large field of view of BACARDI in high altitude (Jäkel et al., 2013), the mean surface albedo might be reduced compared to a $100\,\%$ closed sea ice cover. A direct comparison between the IFS climatological spectral sea ice albedo and the measured one is not possible as the BACARDI measurements can only be used to derive the broadband albedo. However, the comparison of the broadband albedo also needs to consider that the BACARDI measurements provide a sea ice albedo for cloudy diffuse illumination conditions, which the IFS sea ice albedo parameterization does account for but uses constant values for the diffuse albedo. The IFS predicts a nearly constant broadband albedo of 0.76 for both RF 17 and RF 18, whereas the BACARDI derived below-cloud broadband albedo ranges from 0.6 to 0.89 for RF 17 and 0.71 to 0.93 for RF 18. The observed mean albedo value for RF 17 (0.72) is thus lower than the albedo assumed in the IFS, while for RF 18 (0.82) the albedo is underestimated by the IFS.

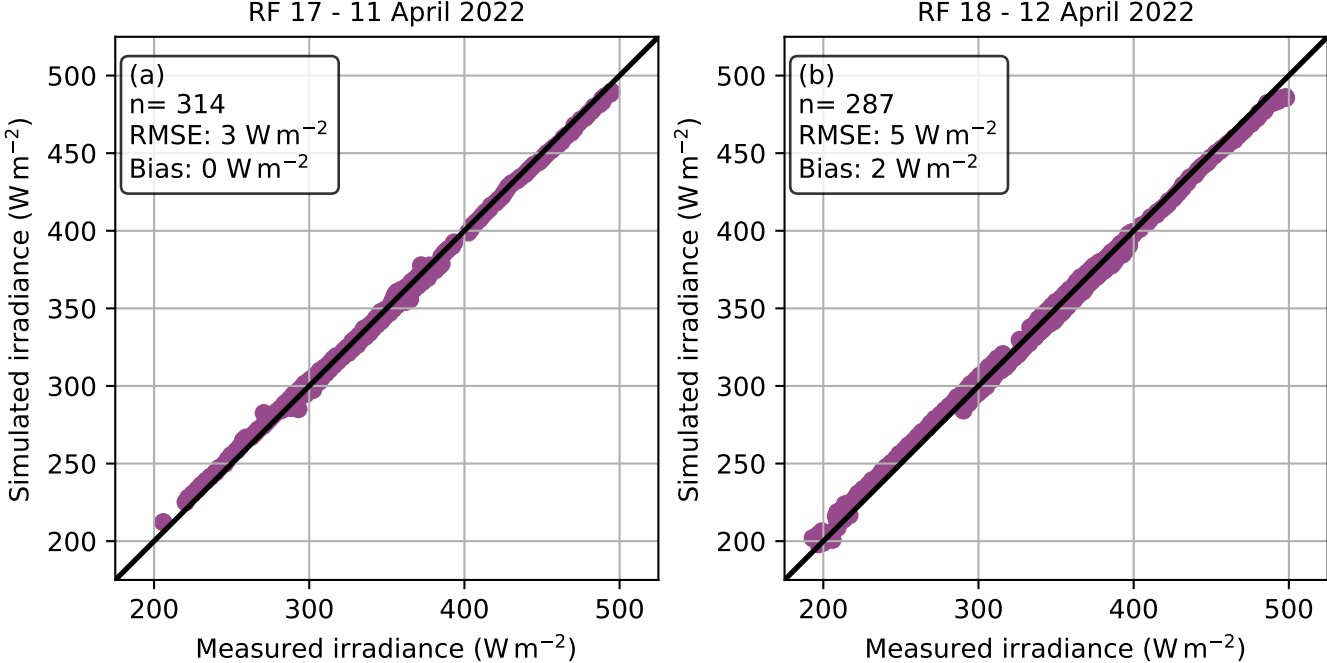

**Figure 6.** Scatter plot of minutely mean BACARDI measured diffuse solar downward irradiance against ecRad simulations for times when HALO was above 11 km altitude for (a) RF 17 and (b) RF 18. The 1:1 line is shown in black.

To investigate the influence of this mismatch, two simulations are performed, in which the spectral albedo provided by the parameterization from Ebert and Curry (1993) is scaled with the measured broadband albedo from BACARDI. For these simulations the Fu-IFS ice optics parameterization is used. To compare the reference simulation with the experiment the mean solar transmissivity below cloud is used. Here, the solar transmissivity is calculated from the downward irradiance above-cloud derived from simulations by ecRad and the below-cloud measurements of either BACARDI or the below-cloud simulations at flight level. The solar transmissivity, as a relative measure, thereby mostly compensates for the effect of the solar zenith angle, which would otherwise dominate the measurement. The mean solar transmissivity below cloud is reduced from $0.78$ in the reference simulation to $0.77$ for RF 17 and from $0.56$ to $0.54$ for RF 18 (see Table 3). As this change is minute and also in the wrong direction the sea ice albedo representation in the IFS does not seem to be the major problem for these cases.

## 5.2 Ice optics parameterizations

To investigate the influence of the three chosen ice optics parameterization (see Table 2) two additional simulations are performed, in which only the ice optics parameterization is changed from the operational Fu-IFS to Yi2013 or Baran2016. The difference between those three simulations and the BACARDI measurements is best represented using probability density functions (PDFs) of the below-cloud section. Again the solar transmissivity is used to compare the simulations and measurements.

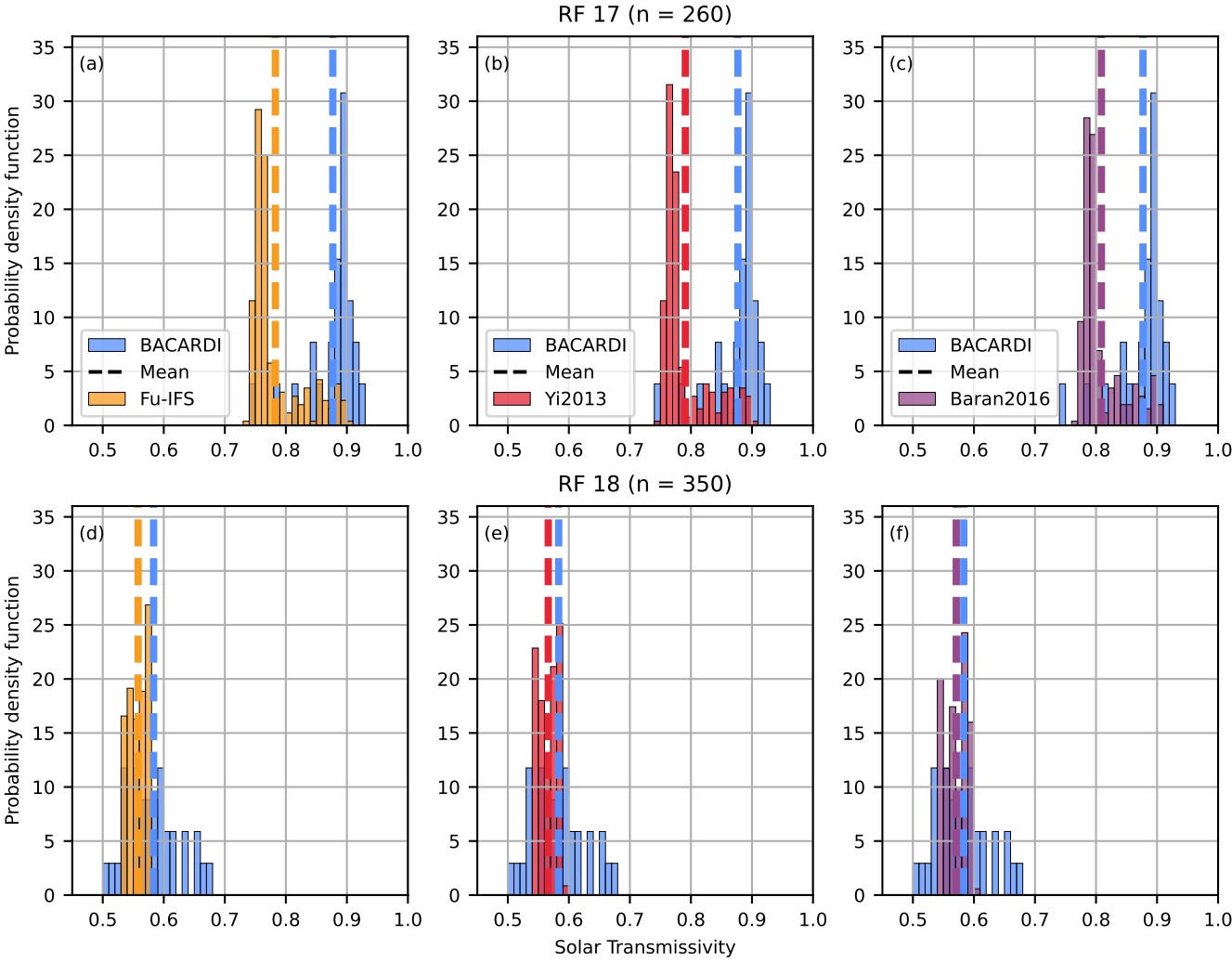

**Figure 7.** Histograms of solar transmissivity below cloud with bin size of 0.01 for (a - c) RF 17 and (d - f) RF 18. Each column compares the minutely averaged below-cloud BACARDI measurements with a simulation using a different ice optics parameterization: (a, d) Fu-IFS, (b, e) Yi2013 and (c, f) Baran2016. The number of samples n is shown in the row title.

Figure 7 shows the distributions of solar transmissivity below cloud from the ecRad simulations compared to the temporally averaged BACARDI measurements with the mean depicted as the dashed line. RF 17 is characterized by higher transmissivities (mean value of 0.88) compared to RF 18 (mean value of 0.58). This is in line with the geometrically and optically thin cirrus observed in RF 17 compared to the geometrically and optically thicker cirrus of RF 18. However, for RF 17 the simulations show a mean transmissivity of 0.78 and do not reproduce this high transmissivity of the measurements. Switching the ice optics parameterization to either Yi2013 or Baran2016 improves the agreement slightly by increasing the mean values to 0.79 and 0.81 respectively. The same holds for RF 18 but the differences between the simulations and measurements are below 0.02. Another difference is the width of the distribution, which does not match as well for RF 18 as for RF 17. Thus, although the mean transmissivity matches better, the simulations do not represent the whole width of the measured range for RF 18. The choice of ice optics parameterization can, therefore, not fully explain the difference between observations and the reference IFS-ecRad simulation (Fu-IFS).

## 5.3 Ice effective radius parameterization

The cirrus radiative properties are determined by the IWC, the ice water path (IWP), which is the integral of the IWC over height, and $r_{\text{eff}}$. However, as introduced above only the IWC is a prognostic variable while $r_{\text{eff}}$ needs to be parameterized. Thus, another source of uncertainty in the ecRad simulations stems from this parameterization. To explore the sensitivity of the ecRad simulations to $r_{\text{eff}}$ the $12\,\text{UTC}$ time step from the below cloud section of RF 18 is taken and $r_{\text{eff}}$ inside the cloud is varied in the possible range of values between $13\,\mu\text{m}$ and $100\,\mu\text{m}$. The simulations are preformed using the reference setup with the IFS IWC and the Fu-IFS ice optics parameterization. Comparing the solar downward irradiance below the cloud across all simulations shows a change between $-5\,\%$ to $+35\,\%$ with respect to the original value. Thus, we conclude that $r_{\text{eff}}$ is indeed one of the driving factors in the ecRad simulations. This poses the question which input parameter of the Sun (2001) parameterization are most crucial to improve the parameterization for the Arctic?

An important feature added during the implementation of the $r_{\text{eff}}$ parameterization in the IFS was to scale the minimum $r_{\text{eff}}$ with the cosine of latitude. Thus, smaller $r_{\text{eff}}$ are possible in the high latitudes compared to the tropics. Recent in situ observations showed larger ice crystals also in high latitudes (De La Torre Castro et al., 2023) and suggest that the extrapolation of this cosine dependency might be misleading. Removing this cosine dependency, therefore, sets a higher lower bound for the predicted $r_{\text{eff}}$ in the Arctic. The lower bound for the simulations in the case study regions is $13\,\mu\text{m}$. Without the cosine dependency the lower bound is lifted to $39\,\mu\text{m}$. Thus, all $r_{\text{eff}}$ values below this value are set to the new lower bound.

As $r_{\text{eff}}$ depends on the IWC, another experiment is set up, in which the retrieved IWC from VarCloud together with the temperature of the IFS are used as input to the Sun (2001) parameterization. This setup can be further varied by also turning the cosine dependency of the minimum $r_{\text{eff}}$ off. Turning the cosine dependency off leads to a shift of the minimum $r_{\text{eff}}$ towards the new minimum of $39\,\mu\text{m}$. Compared to the original $r_{\text{eff}}$ distributions shown in Fig. 9 (e) and (f), most $r_{\text{eff}}$ values are now in the smallest available bin leading to a heavily right skewed distribution (not shown). The values above $39\,\mu\text{m}$ are not changed. Changing the IWC from IFS to the VarCloud values causes only small differences when the cosine dependency is on. These differences are mostly present at values smaller than the new minimum $r_{\text{eff}}$ and, thus, there is almost no difference between

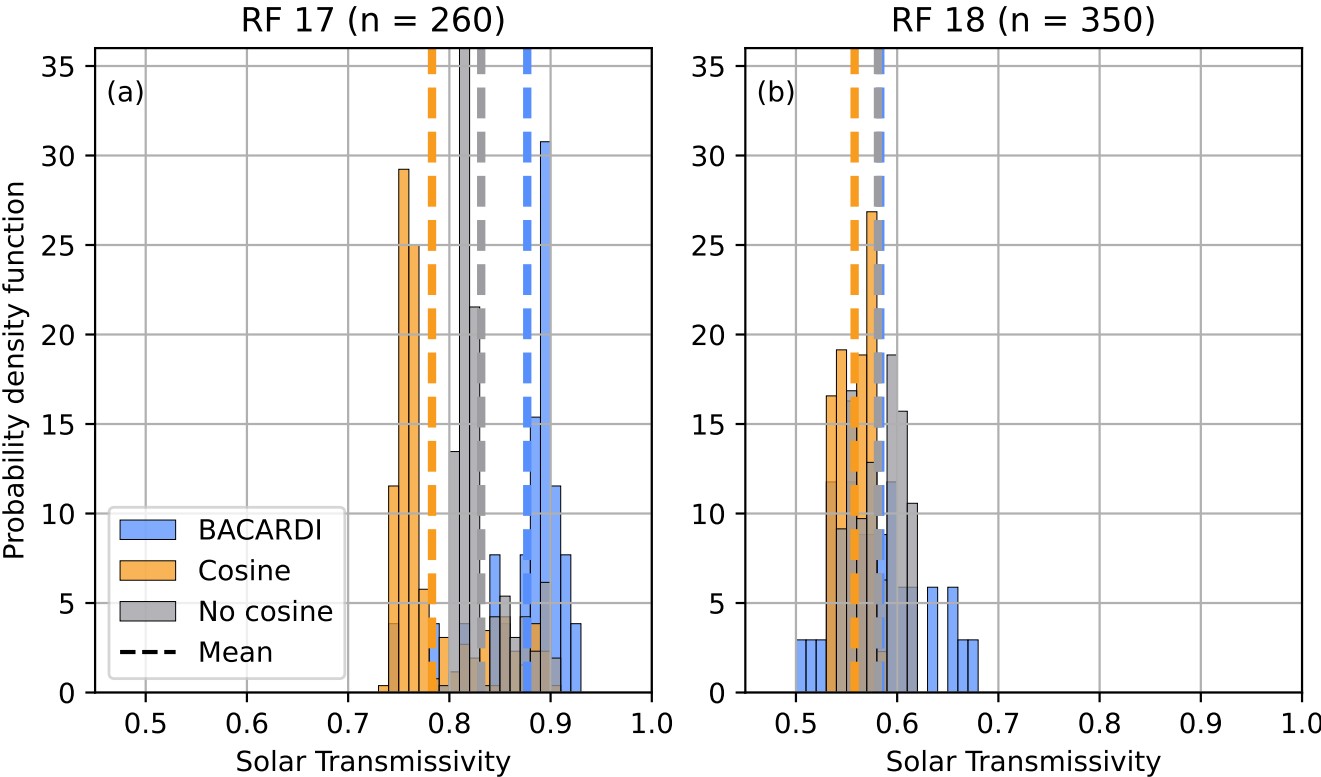

**Figure 8.** Solar transmissivity below cloud as measured by BACARDI and simulated by ecRad using the Fu-IFS ice optics parameterization turning the cosine dependency for the calculation of the minimum $r_{\mathrm{eff}}$ on (Cosine) and off (No cosine) for (a) RF 17 and (b) RF 18.

the distributions when the cosine dependency is off. Following these observations two more simulations are performed using the IFS IWC as input and turning the cosine dependency of the minimum $r_{\mathrm{eff}}$ off.

The results of these simulations are shown in Fig. 8 for (a) RF 17 and (b) RF 18. These simulations use the Fu-IFS ice optics parameterization but the results using Yi2013 differ only slightly and show the same trend (not shown). The "No cosine" simulations show a higher mean solar transmissivity compared to the "Cosine" simulations. For RF 18 this leads to a perfect match of the mean transmissivity with the measured one by BACARDI (see Table 3). However, the spread of the measurements is still not reproduced. RF 17 is still missing the high transmissivity but the mean is noticeably shifted from

$0.78$ to $0.83$ improving the match with the observations. This experiment showed that an improved performance of ecRad for Arctic cirrus can be achieved by removing the cosine dependency in the IFS's implementation of the Sun (2001) $r_{\mathrm{eff}}$ parameterization. However, the cirrus analysed in this study was formed in the Arctic and fulfills the classification of in situ formed cirrus. For cirrus formed via mixed-phase clouds, the conclusion of this study may not hold and smaller ice crystals might be more realistic in this scenario. Thus, a parameterization considering the nature of the cirrus formation might lead to

a more realistic representation of $r_{\mathrm{eff}}$ in the IFS.

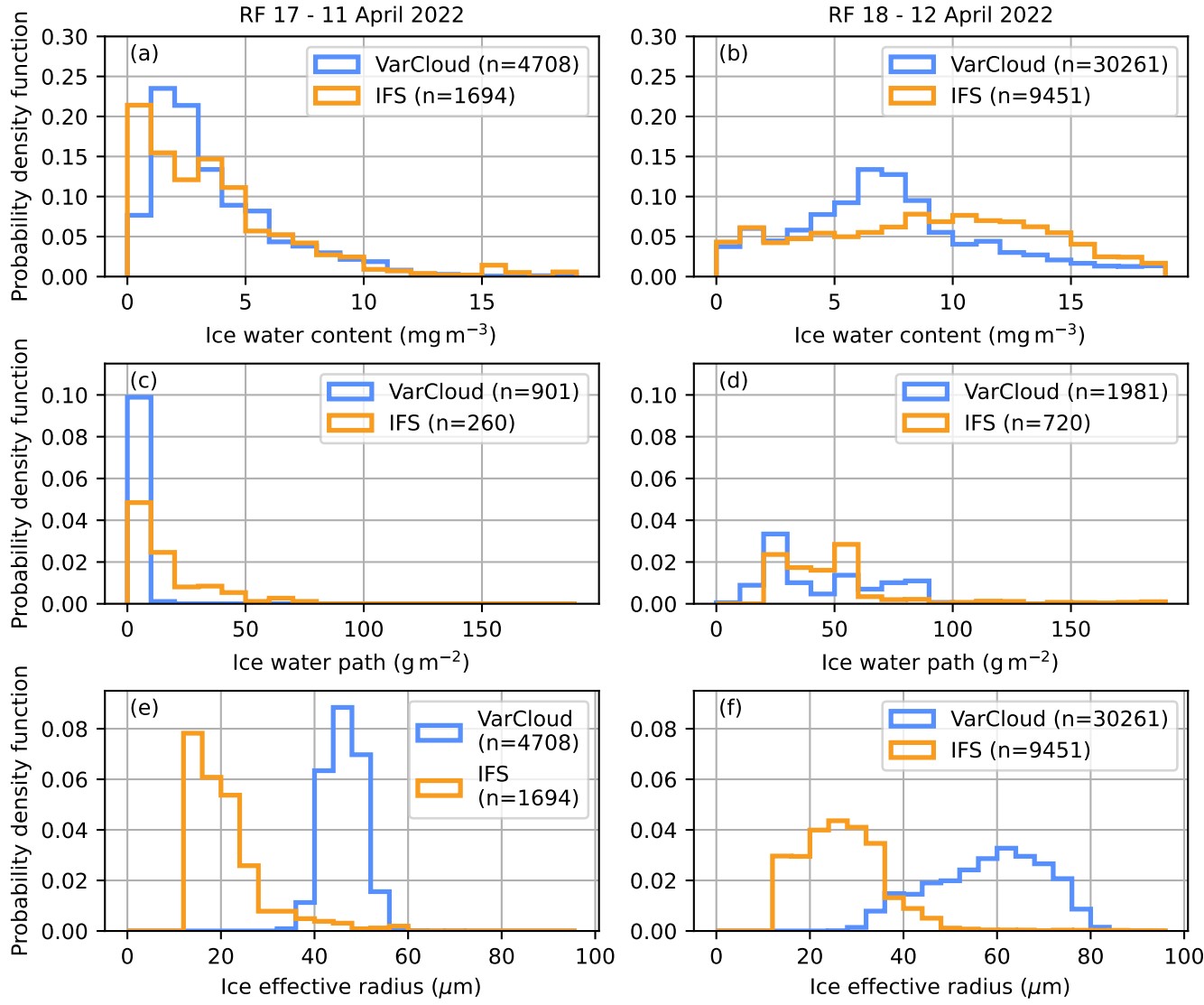

**Figure 9.** Probability density functions of (a, b) IWC with $1\,\mathrm{mg\,m^{-3}}$ binwidth, (c, d) IWP with $10\,\mathrm{g\,m^{-2}}$ binwidth and (e, f) $r_{\mathrm{eff}}$ with $4\,\mathrm{\mu m}$ binwidth for (a, c, e) RF 17 and (b, d, f) RF 18 of the IFS/parameterization output from the below-cloud section and the VarCloud retrieval. n depicts the number of points used in each histogram.

## 5.4  IWC and $r_{\mathrm{eff}}$ input

To evaluate whether the ice optics parameterizations perform better with more realistic values of $r_{\mathrm{eff}}$ and IWC the VarCloud estimated IWC and $r_{\mathrm{eff}}$ are used as independent measures. The probability density functions of the predicted IWC, IWP and $r_{\mathrm{eff}}$ for the below-cloud sections of RF 17 and RF 18 and the retrieved values from the above-cloud sections are shown in

Fig. 9. Due to the temporal resolution of VarCloud, more points are available for the retrieval compared to the IFS while the IWP distributions naturally have less data points. The distributions for the IWC look rather similar with the major difference that the IFS shows more smaller values in the $0 - 1\,\mathrm{mg\,m^{-3}}$ bin for RF 17, whereas the VarCloud data has a peak at the $1 - 2\,\mathrm{mg\,m^{-3}}$ and $2 - 3\,\mathrm{mg\,m^{-3}}$ bin. Both distribution for RF 17 are positively skewed. For RF 18 the distributions are flatter and show more large values than observed in RF 17. The VarCloud values show a flat peak at the $6 - 8\,\mathrm{mg\,m^{-3}}$ bins, whereas

the IFS data do not show a very distinct peak at all. However, the IFS shows more large values. For RF 17 only very low IWP values below $20\,\mathrm{g\,m^{-2}}$ are observed in the VarCloud data, while the IFS also shows values up to $80\,\mathrm{g\,m^{-2}}$. Nonetheless, most of the IFS values are also concentrated in the first two bins below $20\,\mathrm{g\,m^{-2}}$ giving both distributions a strong positive skewness. This shows that, although the IFS predicts more smaller IWC values during RF 17, it overpredicts the IWP and thus the optical thickness as has already been shown with the solar transmissivity in Fig. 7. The histograms for RF 18 in Fig. 9 (d),

in analogy to the IWC distributions, are flatter and show more large values. Apart from the VarCloud data showing more values in the lower bins between $20\,\mathrm{g\,m^{-2}}$ and $40\,\mathrm{g\,m^{-2}}$ and the IFS data having values above $90\,\mathrm{g\,m^{-2}}$, which the VarCloud data is missing, the distributions are rather similar. A more pronounced difference can be observed for $r_{\mathrm{eff}}$ where the Sun (2001) parameterization in the IFS predicts smaller particles with a mean of $21.1\,\mathrm{\mu m}$ for RF 17 and $26.7\,\mathrm{\mu m}$ for RF 18 compared to the retrieved $r_{\mathrm{eff}}$ from VarCloud showing a mean of $46.3\,\mathrm{\mu m}$ for RF 17 and $57.9\,\mathrm{\mu m}$ for RF 18. The shape of the distributions

is hereby similar.

     As $r_{\mathrm{eff}}$ and IWP affect the optical properties of the cirrus, e.g., a smaller $r_{\mathrm{eff}}$ will result in a higher cloud optical thickness for the same IWC, these differences suggest that using the VarCloud retrieval as input could improve the ecRad simulations. To test this hypothesis, a sensitivity study is conducted, in which the IWC and $r_{\mathrm{eff}}$ retrieved by VarCloud are used as input for ecRad simulations of the below-cloud section. For this test, the full temporal resolution of the remote sensing observations is

used, as the test is not restricted to the IFS grid box size. To better compare the simulations, the vertical resolution of VarCloud is interpolated onto the IFS vertical resolution. Since the VarCloud retrieval represents the actual heterogeneity of the cloud, there is no need to introduce any sub-grid variability. Thus, the value of the fractional standard deviation (`fractional_std`) is set to 0. It describes the horizontal variability of the cloud water content in a model layer (ecRad documentation website) and is set to 1 in the IFS and in the simulations shown in the previous section. The VarCloud simulations are also done with the

three available ice optics parameterizations. However, since Baran2016 does not use the $r_{\mathrm{eff}}$ as a direct input only the retrieved IWC is used in these simulations.

     Figure 10 shows the result of the VarCloud-driven ecRad simulations using the three different ice optics parameterization Fu-IFS, Yi2013 and Baran2016 compared to the BACARDI measurements. Compared to the IFS-driven simulations, the mean solar transmissivity is closer to the measured one for RF 17 using the operational Fu-IFS (0.81) and the experimental Yi2013

(0.81) ice optics parameterization. For Baran2016 the cloud is simulated as optically thicker than before, leading to a lower mean transmissivity of 0.77. For RF 18, Fu-IFS and Yi2013 overestimate the solar transmissivity by 0.02, whereas Baran2016 got slightly worse with a mean transmissivity of 0.56 compared to 0.57 before. The main reason for these changes is the larger $r_{\mathrm{eff}}$ retrieved with VarCloud. Larger ice crystals at a similar IWC result in fewer ice crystals, which in turn leads to less scattering and therefore higher transmissivity of the cirrus. Since Baran2016 does not use $r_{\mathrm{eff}}$ as input, the difference to the

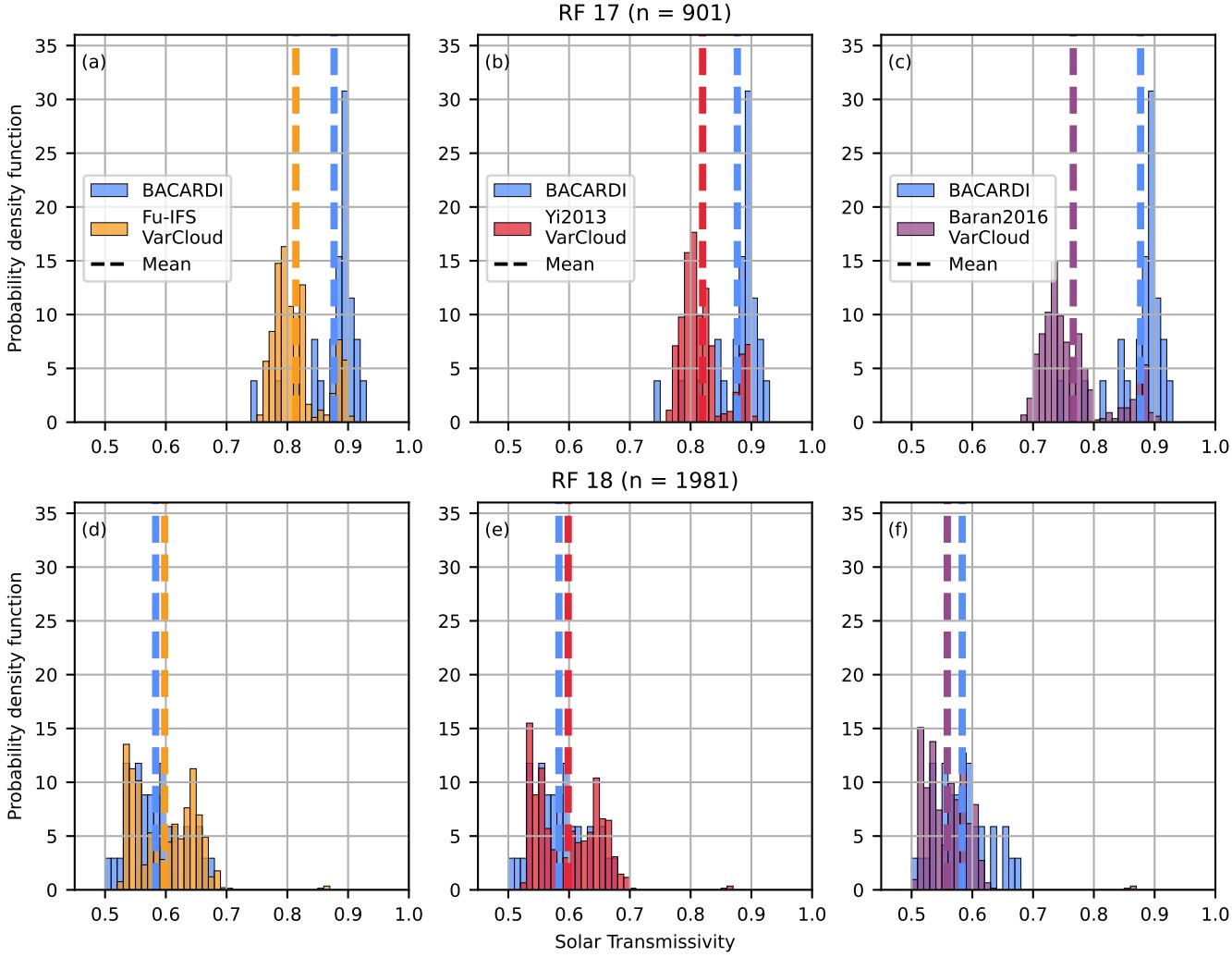

**Figure 10.** As Fig. 7 but using the VarCloud retrieved IWC and $r_{\mathrm{eff}}$ as input for the ecRad simulations. Both BACARDI and ecRad data are calculated for 1 Hz frequency.

IFS-driven simulation is smaller as it is solely caused by the difference in IWC between the IFS and VarCloud. As shown in Fig. 9 this difference is larger for RF 17 with more small values of IWC which in this case leads to an optically thicker cloud.

Table 3 summarizes the mean solar transmissivity calculated with the BACARDI measurements and with the different ecRad setups. It can be seen that, for the IFS-driven simulations, the choice of ice optics parameterization does not change the result significantly. However, with the VarCloud input, the difference between Fu-IFS/Yi2013 and Baran2016 becomes larger. This

indicates that the $r_{\mathrm{eff}}$ parameterization and the underestimation of $r_{\mathrm{eff}}$ by it is the cause for this difference.

**Table 3.** Mean solar transmissivity calculated with the BACARDI measurements (bold) and all ecRad simulations using different input and different ice optics parameterizations. The first row after the BACARDI measurements using the IFS as input and the Fu-IFS ice optics parameterization (italic) is the reference simulation, which is closest to the operational IFS setup.

| Input | Ice Optics Parameterization | Mean Transmissivity | |
|---|---|---|---|
| | | RF 17 | RF 18 |
| **BACARDI** | | **0.88** | **0.58** |
| | Fu-IFS | *0.78* | *0.56* |
| IFS | Yi2013 | 0.79 | 0.57 |
| | Baran2016 | 0.81 | 0.57 |
| IFS scaled sea ice albdeo | Fu-IFS | 0.77 | 0.54 |
| | Fu-IFS | 0.83 | 0.58 |
| IFS No Cosine | Yi2103 | 0.84 | 0.59 |
| | Fu-IFS | 0.81 | 0.60 |
| VarCloud | Yi2013 | 0.82 | 0.60 |
| | Baran2016 | 0.77 | 0.56 |

To better understand why $r_{\mathrm{eff}}$ is so important in these cases the optical depth resulting from the IWC and $r_{\mathrm{eff}}$ is calculated. As the incident wavelength is small compared to the ice crystal size in the solar part of the spectrum, the geometric optic assumption holds and the extinction coefficient $\beta_{\mathrm{ext}}$ can be calculated following Francis et al. (1994) with:

$$\beta_{\mathrm{ext}} = \frac{3}{2} \frac{\mathrm{IWC}}{\rho_{\mathrm{ice}} \cdot r_{\mathrm{eff}}} \qquad (1)$$

with $\rho_{\mathrm{ice}} = 916.7\,\mathrm{kg\,m^{-3}}$ being the density of ice. Vertically integrating $\beta_{\mathrm{ext}}$ over the whole cloud results in the optical depth. This is the primary variable influencing the transmissivity and using the IFS predicted values of IWC and $r_{\mathrm{eff}}$ to calculate it results in a mean optical depth of $0.42$ with a standard deviation of $0.17$ for RF 17 and $1.9$ $(0.36)$ for RF 18. Using the VarCloud retrieved values instead leads to lower optical depths for both cases with $0.27$ $(0.24)$ for RF 17 and $1.14$ $(0.52)$ for RF 18. This explains why the simulations with Fu-IFS and Yi2013 produce higher transmissivities for both cases when using

the VarCloud retrieval as input. However, even though the performance of ecRad is improved for RF 17 some radiative effects

are still missing in the simulation. 3D radiative effects play a more important role at high solar zenith angles and could explain some of the difference. The VarCloud-driven simulations for RF 18 result in an overestimation of solar transmissivity for the Fu-IFS and Yi2013 ice optics parameterization caused by the larger $r_{\text{eff}}$. It has to be mentioned here that also the retrieval is based on assumptions of particle habit and as such is not perfect. Thus, some of the difference can be explained by the retrieval uncertainty of $r_{\text{eff}}$.

## 6 Summary and Conclusions

This study evaluated the ability of the Integrated Forecasting System (IFS) to accurately simulate optical and microphysical properties of Arctic cirrus. Airborne measurements of cirrus transmissivity of two cirrus cases, one optically thin and one optically thick, were compared to offline radiative transfer simulations by ecRad initialized with IFS forecasts. The standard IFS/ecRad configuration showed lower values of cirrus transmissivity compared to the observations for the optically thin cirrus. For optically thick cirrus, only slight differences were observed. We investigated the influence of different ice optics parameterizations available within ecRad, namely the operational one from Fu (1996) (Fu-IFS), and the two experimental ones from Yi et al. (2013) (Yi2013) and Baran et al. (2016) (Baran2016).

Concluding from the results of this case study, a change in ice optics parameterization does not result in better model performance. Instead, cloud microphysical properties were identified to be a possible reason for the mismatch. Replacing both the ice water content (IWC) and ice ffective radius ($r_{\text{eff}}$) with retrieved values based on active remote sensing measurements on HALO improves the match between the measured solar transmissivity and the simulations for RF 17. This holds for operating ecRad with the Fu-IFS and the Yi2013 ice optics parameterization. When using Baran2016, where only the IWC can be replaced with the retrieved values, the differences increased. For the optically thick cirrus of RF 18 this sensitivity test causes an overestimation of the solar transmissivity when using Fu-IFS and Yi2013, whereas it causes a slightly better match for Baran2016. This indicates that the IWC forecasted by the IFS is realistic and the main reason for the mismatch between the simulated and measured solar transmissivities is the $r_{\text{eff}}$ assumption, either given by the parameterization from Sun (2001) or encoded within Baran2016.

Although not significantly improving performance, applying a new ice optics parameterization such as the one from Baran et al. (2016) has other positive side effects. First, the parameterization is based on more recent in situ measurements of ice crystals and uses a parameterization by Field et al. (2007) to generate particle size distributions (PSDs) across a wide range of temperature and IWC values. This improves the statistical basis of the parameterization and makes it more likely to be valid for the Arctic. Second, the Baran2016 parameterization removes the dependence on the $r_{\text{eff}}$ parameterization from Sun (2001). The Sun (2001) parameterization is based on the PSD parameterization developed by McFarquhar and Heymsfield (1997), which they derived from measurements in the Tropics during the Central Equatorial Pacific EXperiment (CEPEX). Thus, the measurement data used may not be representative of the mid-latitudes or the Arctic. Further, to account for the observation that ice crystals measured in the tropics are generally larger than the ones in the mid-latitudes, a cosine weighting depending on the latitude is included in the IFS's definition of the minimum $r_{\text{eff}}$. This makes sure that with increasing latitude the minimum

$r_{\text{eff}}$ decreases. However, recent in situ observations presented by De La Torre Castro et al. (2023) show that the size of ice crystals in Arctic cirrus is on average larger than in the mid-latitudes. The cosine weighting only affects the lower bound of $r_{\text{eff}}$. Nonetheless, due to the IWC predicted by the IFS and the underlying in situ data of the parameterization, larger particle sizes are not predicted under Arctic conditions. Removing the cosine dependency from the Sun (2001) parameterization shows better results and suggests, that further in situ measurements are needed to improve the current parameterization by a more complex latitudinal dependency and account for Arctic cirrus.

RF 17 shows an optically thin cirrus, which was highly transmissive for solar radiation. This transmissivity could not be reproduced with the simulations. Thus, another factor is missing in the simulation that would be able to increase the transmissivity. At such high solar zenith angles, 3D effects become of more importance but are ignored in the operational setup of ecRad. Recent work tries to parameterize these 3D effects within ecRad (Hogan et al., 2019). Simulations with these parameterizations turned on, however, do not yield improved results (not shown).

*Code and data availability.* The basic HALO data (altitude, true air speed, location) and the WALES data are available via the HALO Database. The BACARDI data (https://doi.org/10.1594/PANGAEA.963739, Luebke et al., 2023) and the HAMP data (https://doi.org/10.1594/PANGAEA.963250, Dorff et al., 2023) are available via Pangaea. The dropsonde data is under final quality control and will be published on PANGAEA. The VarCloud retrieval is available upon request. The IFS output used in this study was downloaded directly from the ECMWF servers using the Meteorological Archival and Retrieval System (registration required). The trace gas data used in the ecRad simulations is available at the ecRad documentation website. A repository with a Fortran namelist detailing the ecRad setup, a sample input file and the python code for the figures is available on GitHub, last accessed 17.05.2024.

*Author contributions.* JR performed the analysis, the radiative transfer simulations, and drafted and finalized the manuscript. AE and MW contributed to the conception and the design of the study. HM and RJH contributed to the setup of the radiative transfer simulations. FE performed the VarCloud retrieval and contributed to the design of the radiative transfer simulations. BK performed the trajectory calculations. All authors contributed to the discussion of the results. All authors revised the manuscript.

*Competing interests.* The authors declare no competing interests.

*Acknowledgements.* We thank Kevin Wolf for a fruitful discussion and proof reading the manuscript. We thank the whole HALO–$(\mathcal{AC})^3$ team for making the campaign possible and Geet George for the processing of the dropsonde data. We thank Matthew A. Petroff for providing freely available scientific colormaps, enhancing the visual quality of this work (Petroff, 2021). Parts of the results in this work make use of the colormaps in the CMasher package (Van Der Velden, 2020). We acknowledge the use of imagery provided by services from NASA's Global Imagery Browse Services (GIBS), part of NASA's Earth Observing System Data and Information System (EOSDIS). We gratefully acknowl-

edge the funding by the Deutsche Forschungsgemeinschaft (DFG, German Research Foundation) – Project Number 268020496 – TRR 172, within the Transregional Collaborative Research Center "ArctiC Amplification: Climate Relevant Atmospheric and SurfaCe Processes, and Feedback Mechanisms $(\mathcal{AC})^3$". We are further grateful for funding of project grant no. 316646266 by the Deutsche Forschungsgemeinschaft (DFG, German Research Foundation) within the framework of Priority Programme SPP 1294 to promote research with HALO.

500

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
