# Peer review of "Evaluating the Representation of Arctic Cirrus Solar Radiative Effects in the IFS with Airborne Measurements"

_EGUsphere, 2024_

## Referee Comment (RC1)

**Review of 'Evaluating the representation of Arctic Cirrus Solar Radiative Effects in the IFS with Airborne Measurements ' by Röttenbacher et al.**

This study leverages the HALO campaign airborne dataset to evaluate the representation of the solar radiative effects of Arctic cirrus clouds during two case studies in the Integrated Forecast System.
The ecRad radiative transfer scheme is used and sensitivity tests are performed regarding the choice of the ice optics parameterization.
After the evaluation of radiative fluxes and ice crystal properties with respect to radiation observation and lidar-radar retrievals, the paper concludes that the discrepancies between simulaed and observed irradiances are mainly due to the mismatch between observed and simulated ice crystal effective radius.

The paper is overall well written, the methodology is sound, the analysis careful and accurate and the results relevant for global and polar climate modelers. I think the paper can be published in ACP after some revision work following suggestions below.

**Major comments :**

- I am missing something at the end of the paper regarding the conclusions about the reff parameterization, and more particularly the Sun et al. parameterization.
  The paper provides a lot of context in the Introduction (line 61-80) and clearly shows that reff is the culprit of the story. However, as a polar climate modeler, I would be happy to have a suggestion on how to change the original parameterization to make it more 'arctic suitable'. Even though deriving a new reff parameterization is not the main aim of the study, and even though the paper considers only two study cases, I sincerely think this aspect should be tackled (at least a first try) in the paper.
  I would suggest the authors to complement their study with an additional section discussing more in details the performance of - and possible adaptations to - the Sun et al. parameterization for reff. This section might include :
  - an assessment of the reff prediction removing the cosine dependency upon latitude ;
  - a comparison between observed and predicted (by the parameterization) reff values using the observed temperature and IWC as inputs ;
  - a derivation and evaluation of a new reff=f(iwc,T) function from in situ data and a comparison with the original parameterization

- Although they are optically quite thin, I would really appreciate to see a satellite image (infrared and/or visible channels) of the two cirrus clouds studied. This would make it possible to better characterize the horizontal size of the clouds as well as to better visualize which part of the cloud have been sampled by dropsondes. This is quite important since IFS fails in capturing the supersaturation within the cloud.

**Minor comments :**

l26 : 'exhibits specific dependencies on the high gradients of surface albedo' : not clear, please rephrase.

L141 : 'ecRad cloud free simulations' : please provide more details about the simulations setup.

L160 : 'to parameterize 3D radiative effects' : all 3D radiative effects or only trapping ?

L163 : So what is ths spectral resolution ?

L169 : length scale : horizontal resolution ?

L198 : Can you be more explicit on which albedo value is used for each of the ecRad band ?

L206 : Although you mention them in the Introduction, please recall here the output quantities of the ice optics parameterization.

L265 : Could this be due to the fact that IFS does not predict a cloud fraction associated with precipitating ice. What I mean here is that can the model simulates precipitating ice (snow category) in meshes where cloud fraction is 0 ?

L415 : Please provide Pangaea links for BACARDI, HAMP and dropsonde data.

---

## Author Comment (AC1)

**Reply to Review 1**

*"This study leverages the HALO campaign airborne dataset to evaluate the representation of the solar radiative effects of Arctic cirrus clouds during two case studies in the Integrated Forecast System. The ecRad radiative transfer scheme is used and sensitivity tests are performed regarding the choice of the ice optics parameterization. After the evaluation of radiative fluxes and ice crystal properties with respect to radiation observation and lidar-radar retrievals, the paper concludes that the discrepancies between simulated and observed irradiances are mainly due to the mismatch between observed and simulated ice crystal effective radius. The paper is overall well written, the methodology is sound, the analysis careful and accurate and the results relevant for global and polar climate modelers. I think the paper can be published in ACP after some revision work following suggestions below."*

Thank you very much for the positive and critical review of our paper. Before we address your comments below, we would like to mention a few changes in the revised manuscript.

- In previous simulations using the VarCloud data as input we did not replace the cloud fraction from the IFS. We have now included this replacement and updated the solar transmissivity in the text as well as in the revised Fig. 9.
- Due to a comment by reviewer 2 we added two new panels (c) and (d) to Fig. 9 showing the IWP distributions from the IFS and VarCloud for the case study areas in RF 17 and RF 18, respectively.
- In the original version of Fig. 3 the abscissa axis in panel (d) was reversed, which it should not since RF 18 featured a circular flight pattern. We corrected this and the data is now displayed at the correct location.
- Figure 4: A new version of the dropsonde data set became available since submission of the manuscript. We updated Fig. 4 with this revised data set. The main difference is the removal of the NAN values visible in the previous relative humidity plots (panels (b) and (d)) and the removal of data above 10 km altitude, which is due to the interpolation of the data onto a uniform altitude grid.

We also polished the text in the Abstract, the Introduction and the Summary and Conclusions. The rest of the reply is structured as follows, we first repeat your comment in blue italics and then reply to it. We then quote the introduced changes in italics giving the line numbers in the revised manuscript. The revised figures can be found at the end of the reply. Within the reply we use the same abbreviations as in the manuscript, namely

- HALO (High Altitude LOng range research aircraft)
- IFS (Integrated Forecasting System)
- $r_{\text{eff}}$ (ice effective radius)
- IWC (ice water content)
- IWP (ice water path)
- VarCloud (referring to the VarCloud microphysical retrieval from Ewald et al. (2021))
- ecRad (referring to the radiative transfer scheme (Hogan and Bozzo 2018))
- Fu-IFS, Yi2013 and Baran2016 (referring to the ice optics parameterization from Fu (1996), Yi et al. (2013) and Baran et al. (2016), respectively)

**Major comment 1**

*"I am missing something at the end of the paper regarding the conclusions about the $r_{\text{eff}}$ parameterization, and more particularly the Sun et al. parameterization. The paper provides a lot of context in the Introduction (line 61-80) and clearly shows that $r_{\text{eff}}$ is the culprit of the story. However, as a polar climate modeler, I would be happy to have a suggestion on how to change the original parameterization to make it more 'arctic suitable'. Even though deriving a new $r_{\text{eff}}$ parameterization is not the main aim of the study, and even though the paper considers only two study cases, I sincerely think this aspect should be tackled (at least a first try) in the paper. I would suggest the authors to complement their study with an additional section discussing*

*more in details the performance of - and possible adaptations to - the Sun et al. parameterization for $r_{\text{eff}}$. This section might include :"*

Thanks to these detailed suggestions we considered adding a full new section on potential improvements of the parameterization but finally decided to merge it into the available paper structure. Reason is, that the limited observational data (no in situ measurements, only two cases, see detailed reply below) does not allow us to draft a new parameterization. However, in the revised manuscript, we tried to shine some light on the questions: What potentially needs to be changed and what approximation is most crucial? For this purpose, we split the former Sect. 4 into two new sections with the revised Sect. 4 focusing on the comparison between the IFS forecast and the measured macrophysical properties of the cirrus (formerly Sect. 4.1), while the revised Sect. 5 includes a new subsection 5.2 focusing on the $r_{\text{eff}}$ parameterization. Further, we moved the sea ice albedo section (formerly Sect. 4.2.3) to Sect. 5.1 to follow the same structure as in Sect. 3, where we also start with the sea ice albedo. The results of this experiment are also added to Table 3 showing the mean solar transmissivity of all conducted experiments. Due to this change in structure, we also moved the explanation of the calculation of the solar transmissivity into Sect. 5.1.

**L319-L327**: *"To investigate the influence of this mismatch, two simulations are performed, in which the spectral albedo provided by the parameterization from Ebert and Curry (1993) is scaled with the measured broadband albedo from BACARDI. For these simulations the Fu-IFS ice optics parameterization is used. To compare the reference simulation with the experiment the mean solar transmissivity below cloud is used. Here, the solar transmissivity is calculated from the downward irradiance above-cloud derived from simulations by ecRad and the below-cloud measurements of either BACARDI or the below-cloud simulations at flight level. The solar transmissivity, as a relative measure, thereby mostly compensates for the effect of the solar zenith angle, which would otherwise dominate the measurement. The mean solar transmissivity below cloud is reduced from 0.78 in the reference simulation to 0.77 for RF 17 and from 0.56 to 0.54 for RF 18 (see Table 3). As this change is minute and also in the wrong direction the sea ice albedo representation in the IFS does not seem to be the major problem for these cases."*

*"- an assessment of the $r_{\text{eff}}$ prediction removing the cosine dependency upon latitude;*

*- a comparison between observed and predicted (by the parameterization) $r_{\text{eff}}$ values using the observed temperature and IWC as inputs;"*

These are indeed valid points. We address this in the new Sect. 5.2, where we conduct a sensitivity study, in which we turn off the cosine weighting with latitude in the IFS's implementation of the Sun (2001) parameterization. For the second point, we added a description of the changes, which happen to the $r_{\text{eff}}$ distributions showed in Fig. 9 when using the IFS IWC or the VarCloud IWC as input to the Sun (2001) parameterization and turning the cosine weighting on or off. These $r_{\text{eff}}$ values are used in ecRad simulations to analyse the impact on the solar transmissivities. The results are shown in Fig. 8 for the Fu-IFS ice optics parameterization and using the IFS IWC only. The mean solar transmissivity from these simulations is added to Table 3, together with the one simulated when using the Yi2013 ice optics parameterization. A simulation using Baran2016 is not included because the Sun (2001) parameterization is not used in this case. By removing the cosine dependency the simulated solar transmissivity increases and better matches the measured transmissivity. For RF 18 a perfect match of the mean solar transmissivity can be achieved.

**L355-L380**: *"An important feature added during the implementation of the $r_{\text{eff}}$ parameterization in the IFS was to scale the minimum $r_{\text{eff}}$ with the cosine of latitude. Thus, smaller $r_{\text{eff}}$ are possible in the high latitudes compared to the tropics. Recent in situ observations showed larger ice crystals also in high latitudes (De La Torre Castro et al. 2023) and suggest that the extrapolation of this cosine dependency might be misleading. Removing this cosine dependency, therefore, sets a higher lower bound for the predicted $r_{\text{eff}}$ in the Arctic. The lower bound for the simulations in the case study regions is $13\,\mu m$. Without the cosine dependency the lower bound is lifted to $39\,\mu m$. Thus, all $r_{\text{eff}}$ values below this value are set to the new lower bound.*

*As $r_{\text{eff}}$ depends on the IWC, another experiment is set up, in which the retrieved IWC from VarCloud together with the temperature of the IFS are used as input to the Sun (2001) parameterization. This setup can be further varied by also turning the cosine dependency of the minimum $r_{\text{eff}}$ off. Turning the cosine dependency off leads to a shift of the minimum $r_{\text{eff}}$ towards the new minimum of $39\,\mu m$. Compared to the original $r_{\text{eff}}$ distributions shown in Fig. 9 (e) and (f), most $r_{\text{eff}}$ values are now in the smallest available bin leading to a*

*heavily right skewed distribution (not shown). The values above* 39 µm *are not changed. Changing the IWC from IFS to the VarCloud values causes only small differences when the cosine dependency is on. These differences are mostly present at values smaller than the new minimum* $r_{\text{eff}}$ *and, thus, there is almost no difference between the distributions when the cosine dependency is off. Following these observations two more simulations are performed using the IFS IWC as input and turning the cosine dependency of the minimum* $r_{\text{eff}}$ *off.*

*The results of these simulations are shown in Fig. 8 for (a) RF 17 and (b) RF 18. These simulations use the Fu-IFS ice optics parameterization but the results using Yi2013 differ only slightly and show the same trend (not shown). The "No cosine" simulations show a higher mean solar transmissivity compared to the "Cosine" simulations. For RF 18 this leads to a perfect match of the mean transmissivity with the measured one by BACARDI (see Table 3). However, the spread of the measurements is still not reproduced. RF 17 is still missing the high transmissivity but the mean is noticeably shifted from* 0.78 *to* 0.83 *improving the match with the observations. This experiment showed that an improved performance of ecRad for Arctic cirrus can be achieved by removing the cosine dependency in the IFS's implementation of the Sun (2001)* $r_{\text{eff}}$ *parameterization. However, the cirrus analysed in this study was formed in the Arctic and fulfills the classification of in situ formed cirrus. For cirrus formed via mixed-phase clouds, the conclusion of this study may not hold and smaller ice crystals might be more realistic in this scenario. Thus, a parameterization considering the nature of the cirrus formation might lead to a more realistic representation of* $r_{\text{eff}}$ *in the IFS."*

We have added a sentence to Sect. 6 "Summary and Conclusion" on the results of the assessment.

**L474-L476**: *"Removing the cosine dependency from the Sun (2001) parameterization shows better results and suggests, that further in situ measurements are needed to improve the current parameterization by a more complex latitudinal dependency and account for Arctic cirrus.''*

*"- a derivation and evaluation of a new* $r_{\text{eff}}$ *=f(IWC,T) function from in situ data;"*

Unfortunately, we do not have in situ measurements of the two cirrus cases available, as the focus of the HALO–$(\mathcal{AC})^3$ campaign was on remote sensing observations of clouds. The "measured" $r_{\text{eff}}$ and IWC values we present in our paper are derived from the VarCloud retrieval based on radar and lidar remote sensing. It has to be noted, that this retrieval also uses several assumptions and might not show the truth. Building a parameterization on the retrieved $r_{\text{eff}}$ might therefore not be a good idea. Furthermore, we only have two profile measurements of the in-cloud temperature from dropsondes, which are not closely collocated with the radar and lidar measurements. However, this would need to be the case if we tried to find a correlation between the retrieved $r_{\text{eff}}$ and IWC, and the measured temperature.

*"- and a comparison with the original parameterization"*

Although, we did not derive a new parameterization, the sensitivity studies, see replies above, are compared to the original $r_{\text{eff}}$ parameterization from Sun (2001).

**Major comment 2**

*"Although they are optically quite thin, I would really appreciate to see a satellite image (infrared and/or visible channels) of the two cirrus clouds studied. This would make it possible to better characterize the horizontal size of the clouds as well as to better visualize which part of the cloud have been sampled by dropsondes. This is quite important since IFS fails in capturing the supersaturation within the cloud."*

We included a MODIS false color corrected reflectance image for each research flight in Fig. 1 and included a description and interpretation of them in Sect. 2.2 introducing the two case studies.

**L120-L123**: *"In addition to the IFS forecasts Fig. 1 (c) and (d) show the false color corrected reflectance product from the Moderate Resolution Imaging Spectroradiometer (MODIS) on the Terra satellite using Band 3, 6 and 7. This band combination is sensitive to ice and snow and allows to distinguish cirrus, visible as white to slightly orange filaments, from sea ice, which appears in dark orange."*

**L135-L143**: *"This large cirrus field can also be seen in the satellite product depicted in Fig. 1 (c) and (d), which is a combination of overflights from the Terra satellite between 14 UTC and 20 UTC on the respective case study date. In Fig. 1 (c), depicting the situation during RF 17, the edge of the cirrus field can be seen close to the radiosonde dropped at 10 : 42 UTC. Here the cirrus is optically thin while further west on the flight track the optical thickness increases. As indicated by the satellite image, the observations took place at the edge of the cirrus field, which is stretching southwards east of the Greenland coast similar to the IFS forecast. This cirrus field persisted and can be seen again on the 12 April 2022 in Fig. 1 (d). Down to 86° N the cirrus field is rather compact and part of the same air mass. Bigger sections of very optically thick cirrus only appear further south and reach all the way to the sea ice edge close to the radiosonde dropped at 09 : 39 UTC."*

**Minor comments**

For the minor comments we did not adjust the line numbers in your comments. Thus, they still refer to the old version of the manuscript while ours refer to the revised version.

*"L26 : 'exhibits specific dependencies on the high gradients of surface albedo' : not clear, please rephrase."*

**L27-L29**: Rephrased to: *"In contrast to tropic and mid-latitude cirrus, the radiative effect of Arctic cirrus, which we define to occur north of the Arctic circle at 66° N, is strongly influenced by the bimodality of the surface albedo (open ocean vs. sea ice) and the usually low sun."*

*"L141 : 'ecRad cloud free simulations' : please provide more details about the simulations setup."*

**L151-L153**: Rephrased to: *"The transmissivity of the cloud is calculated as the ratio between the below-cloud measurements and the cloud-free downward irradiance at around 11 km, provided by the ecRad simulations described in Sect. 3."*

*"L160 : 'to parameterize 3D radiative effects' : all 3D radiative effects or only trapping ?"*

**L172-L173**: Added *". . . including the radiative transfer through cloud sides and entrapment Hogan et al. (2019)."* to clarify.

*"L163 : So what is this spectral resolution ?"*

We added a new Table (Table 1) with the exact boundaries of the 14 solar bands.

**L180-L181**: *"Thus, the irradiance is calculated for 14 solar bands listed in Table 1."* to clarify the bands of the RRTMG.

*"L169 : length scale : horizontal resolution ?"*

**L182**: Rephrased to *"horizontal resolution"* for consistency and clarity.

*"L198 : Can you be more explicit on which albedo value is used for each of the ecRad band ?"*

We added an explanation on the exact method of how the sea ice surface albedo is calculated. The mean sea ice surface albedo values for the case study sections are further given in Table 1 together with the RRTMG solar bands.

**L211-L214**: *"Linear interpolation is performed in time, treating each of the twelve monthly means as the instantaneous value at the 15th day of each month. These interpolated values are then internally mapped by ecRad to the 14 solar bands defined by the RRTMG using a weighted average according to the overlap of the six albedo bands with the RRTMG bands."*

**L240-L241**: *"The resulting mean solar surface albedo for each RRTMG solar band for the case study period of RF 17 and RF 18 are given in Table 1."*

*"L206 : Although you mention them in the Introduction, please recall here the output quantities of the ice optics parameterization."*

**L230**: Replaced *"bulk optical properties"* with *"extinction coefficient, single-scattering albedo and asymmetry parameter"*.

*"L265 : Could this be due to the fact that IFS does not predict a cloud fraction associated with precipitating ice. What I mean here is that can the model simulates precipitating ice (snow category) in meshes where cloud fraction is 0 ?"*

Yes, the IFS predicts snow/ice even if the cloud fraction is 0. However, this is not accounted for in the radiative transfer simulation as only clouds in levels with a cloud fraction $> 0$ are considered. We added two sentences explaining this.

**L284-L287**: *"It should be mentioned that the IFS does predict small cloud snow water content values (precipitating ice) well below the radar and lidar mask - for RF 18 even down to the surface - yet the important variable for ecRad is the cloud fraction. If no cloud fraction is predicted in a grid layer no cloud optical properties are computed."*

*"L415 : Please provide Pangaea links for BACARDI, HAMP and dropsonde data"*

We added links for BACARDI and HAMP to the Data Availability section. The dropsonde data is in the process of getting a DOI. We hope to include it during the typesetting.

**L483-L484**: *"The BACARDI data (https://doi.org/10.1594/PANGAEA.963739, Luebke et al. 2023) and the HAMP data (https://doi.org/10.1594/PANGAEA.963250, Dorff et al. 2023) are available via Pangaea."*

**Revised figures**

[Figure]

Figure 1: Map of flight tracks with IFS predicted high cloud cover for 12 UTC, sea ice edge (80 % sea ice cover), mean sea level pressure isolines, dropsonde locations (red crosses), highlighted case study regions (orange), and LAGRANTO backward trajectories for (a) RF 17 and (b) RF 18. The box in panel (b) shows a zoom of the case study region with the above and below-cloud flight sections for RF 18. (c) and (d) False color corrected reflectance from MODIS on Terra using Band 3, 6 and 7 for RF 17 and RF 18, respectively, as provided by the Global Imagery Browse Services (GIBS) from NASA.

[Figure]

Figure 3: Measured downward and upward solar irradiance from BACARDI for the (a, b) above and (c, d) below-cloud sections of (a, c) RF 17 and (b, d) RF 18. Panels (e) and (f) show the solar transmissivity below cloud. The x-axis shows the distance traveled by HALO from the start to the end of the above-cloud section.

[Figure]

Figure 4: Atmospheric profiles of (a, c) air temperature and (b, d) relative humidity over ice from the IFS (grey lines) for the whole case study period (above and below-cloud section) along the flight track and the dropsondes (DS) deployed by HALO during the above-cloud section of (a, b) RF 17 and (c, d) RF 18. The black line indicates the flight altitude of HALO during the below-cloud section.

[Figure]

Figure 9: Probability density functions of (a, b) IWC with $1\,\mathrm{mg\,m^{-3}}$ binwidth, (c, d) IWP with $10\,\mathrm{g\,m^{-2}}$ binwidth and (e, f) $r_{\mathrm{eff}}$ with $4\,\mathrm{\mu m}$ binwidth for (a, c, e) RF 17 and (b, d, f) RF 18 of the IFS/parameterization output from the below-cloud section and the VarCloud retrieval. n depicts the number of points used in each histogram.

**References**

Baran, Anthony J., Peter Hill, David Walters, Steven C. Hardiman, Kalli Furtado, Paul R. Field, and James Manners. 2016. "The Impact of Two Coupled Cirrus Microphysics–radiation Parameterizations on the Temperature and Specific Humidity Biases in the Tropical Tropopause Layer in a Climate Model." *J Climate* 29 (14): 52995316. https://doi.org/10.1175/jcli-d-15-0821.1.

De La Torre Castro, Elena, Tina Jurkat-Witschas, Armin Afchine, Volker Grewe, Valerian Hahn, Simon Kirschler, Martina Krämer, et al. 2023. "Differences in Microphysical Properties of Cirrus at High and Mid-Latitudes." *Atmos. Chem. Phys.* 23 (20): 13167–89. https://doi.org/10.5194/acp-23-13167-2023.

Dorff, Henning, Clemantyne Aubry, Florian Ewald, Lutz Hirsch, Friedhelm Jansen, Heike Konow, Mario Mech, et al. 2023. "Unified Airborne Active and Passive Microwave Measurements over Arctic Sea Ice and Ocean During the HALO-(AC)³ Campaign in Spring 2022." https://doi.org/10.1594/PANGAEA.963250.

Ebert, Elizabeth E., and Judith A. Curry. 1993. "An Intermediate One-Dimensional Thermodynamic Sea Ice Model for Investigating Ice-Atmosphere Interactions." *Journal of Geophysical Research: Oceans* 98 (C6): 10085–109. https://doi.org/10.1029/93JC00656.

Ewald, Florian, Silke Groß, Martin Wirth, Julien Delanoë, Stuart Fox, and Bernhard Mayer. 2021. "Why We Need Radar, Lidar, and Solar Radiance Observations to Constrain Ice Cloud Microphysics." *Atmospheric Measurement Techniques* 14 (7): 5029–47. https://doi.org/10.5194/amt-14-5029-2021.

Fu, Qiang. 1996. "An Accurate Parameterization of the Solar Radiative Properties of Cirrus Clouds for Climate Models." *J Climate* 9 (9): 20582082. https://doi.org/10.1175/1520-0442(1996)009%3C2058:aapots%3E2.0.co;2.

Hogan, Robin J., and Alessio Bozzo. 2018. "A Flexible and Efficient Radiation Scheme for the ECMWF Model." *J. Adv. Model. Earth Syst.* 10 (8): 19902008. https://doi.org/10.1029/2018ms001364.

Hogan, Robin J., Mark D. Fielding, Howard W. Barker, Najda Villefranque, and Sophia A. K. Schäfer. 2019. "Entrapment: An Important Mechanism to Explain the Shortwave 3D Radiative Effect of Clouds." *J. Atmos. Sci.* 2019 (1): 4866. https://doi.org/10.1175/JAS-D-18-0366.1.

Luebke, Anna E., André Ehrlich, Johannes Röttenbacher, Martin Zöger, Andreas Giez, Vladyslav Nenakhov, Christian Mallaun, and Manfred Wendisch. 2023. "Broadband Solar and Terrestrial, Upward and Downward Irradiance Measured by BACARDI on HALO During the HALO-(AC)³ Field Campaign in 2022." https://doi.org/10.1594/PANGAEA.963739.

Sun, Zhian. 2001. "Reply to Comments by Greg m. McFarquhar on 'Parametrization of Effective Sizes of Cirrus-Cloud Particles and Its Verification Against Observations.' (October b, 1999,125, 3037–3055)." *Quart. J. Roy. Meteorol. Soc.* 127 (571): 267271. https://doi.org/10.1002/qj.49712757116.

Yi, Bingqi, Ping Yang, Bryan A. Baum, Tristan L'Ecuyer, Lazaros Oreopoulos, Eli J. Mlawer, Andrew J. Heymsfield, and Kuo-Nan Liou. 2013. "Influence of Ice Particle Surface Roughening on the Global Cloud Radiative Effect." *Journal of the Atmospheric Sciences* 70 (9): 2794–2807. https://doi.org/10.1175/JAS-D-13-020.1.

---

## Author Comment (AC2)

**Reply to Review 2**

*"In this manuscript, the authors evaluate the SW-domain properties of Arctic cirrus clouds as simulated by the IFS model by comparing them with airborne measurements. A similar study was recently published in ACP but for low-level clouds. The subject is important and useful. The work is of very good quality, the results are clear, the analyses relevant and the writing pleasant. This manuscript deserves to be published in ACP and only minor comments and suggestions are made below."*

Thank you very much for this positive review of our paper. Before we address your comments and suggestions below, we would like to mention a few changes in the revised manuscript.

- In previous simulations using the VarCloud data as input we did not replace the cloud fraction from the IFS. We have now included this replacement and updated the solar transmissivity in the text as well as in the revised Fig 9.
- In the original version of Fig. 3 the abscissa axis in panel (d) was reversed, which it should not since RF 18 featured a circular flight pattern. We corrected this and the data is now displayed at the correct location.
- Figure 4: A new version of the dropsonde data set became available since submission of the manuscript. We updated Fig. 4 with this revised data set. The main difference is the removal of the NAN values visible in the previous relative humidity plots (panels (b) and (d)) and the removal of data above 10 km altitude, which is due to the interpolation of the data onto a uniform altitude grid.
- Due to comments from reviewer 1, we split the former Sect. 4 into two new sections with the revised Sect. 4 focusing on the comparison between the IFS forecast and the measured macrophysical properties of the cirrus (formerly Sect. 4.1), while the revised Sect. 5 includes a new subsection 5.2 focusing on the $r_{\text{eff}}$ parameterization.
- Further, we moved the sea ice albedo section (formerly Sect. 4.2.3) to Sect. 5.1 to follow the same structure as in Sect. 3, where we also start with the sea ice albedo. The results of this experiment are also added to Table 3 showing the mean solar transmissivity of all conducted experiments. Due to this change in structure, we also moved the explanation of the calculation of the solar transmissivity into Sect. 5.1.

We also polished the text in the Abstract, the Introduction and the Summary and Conclusions. The rest of the reply is structured as follows, we first repeat your comment in blue italics and then reply to it. Please note that the line numbers in your comments are unchanged and still refer to the old version of the manuscript. We then quote the introduced changes in italics giving the line numbers in the revised manuscript. The revised figures can be found at the end of the reply. Within the reply we use the same abbreviations as in the manuscript, namely

- HALO (High Altitude LOng range research aircraft)
- IFS (Integrated Forecasting System)
- $r_{\text{eff}}$ (ice effective radius)
- IWC (ice water content)
- IWP (ice water path)
- VarCloud (referring to the VarCloud microphysical retrieval from Ewald et al. (2021))
- ecRad (referring to the radiative transfer scheme (Hogan and Bozzo 2018))
- Fu-IFS, Yi2013 and Baran2016 (referring to the ice optics parameterization from Fu (1996), Yi et al. (2013) and Baran et al. (2016), respectively)

**Comments**

*"1. l. 273-274: (comment on Fig. 4) If I understand correctly, the measurements shown between 11 and 11:30 correspond to the results of the simulation at 11:00, and the cloud evolution between 11 and 11:30 is*

*due to the displacement of the aircraft (i.e. the spatial evolution of the clouds) and not to the evolution of the clouds over time. I think this should mentioned more explicitly."*

This is indeed correct. The distance HALO flies within 30 min and the covered spatial changes are more substantial than the temporal evolution of the cloud within these 30 min. This is investigated with Fig. 5 showing the IWC distributions for the 11 UTC and 12 UTC time step of the IFS. Apart from a small shift towards smaller values in RF 17 no substantial changes are observed between the two time steps. We adjusted a sentence and added one to clarify this.

**L276-278**: Added *"along the flight track of HALO."* to *"Figure 4 shows the VarCloud lidar-radar cloud mask from HALO, the aircraft altitude, and the predicted cloud fraction of the IFS along the flight track of HALO."*

**L293**: Added *"Thus, the change in cloud fraction shown here is mostly due to HALO flying through different grid cells."*

*"2. l. 291-292: What do you mean by compensate? The effect of the cosine of the zenith angle?"*

The solar zenith angle is quite large for both of our case studies. Especially for RF 17, flying westward, it also changes by about $2°$, which amounts to about $25\,\mathrm{W\,m^{-2}}$ decrease of downward solar irradiance at the top of atmosphere. Thus, the transmitted solar downward irradiance below the cloud is dominated by this change in solar zenith angle. Analyzing the downward irradiance only, makes it harder to interpret other effects such as the inhomogeneity of the cirrus. Thus, we opted for the transmissivity as a relative measure, which partly compensates for changes of the solar zenith angle in our measurements. We rephrased the introductory sentence regarding the solar transmissivity to make this more clear.

**L324-L325**: Rephrased to *"The solar transmissivity, as a relative measure, thereby mostly compensates for the effect of the solar zenith angle, which would otherwise dominate the measurement."*

*"3. Figure 8: What is the value of IWC to differentiate between clear and cloudy skies? How are the histograms modified for small values of IWC when this threshold is changed?"*

This is a good question, with a not so easy answer as the IFS cloud fraction is a prognostic variable itself. Meaning there is no fixed threshold of IWC from which a cloud is diagnosed. It is more so that either there is a cloud fraction value predicted or not. This also allows for precipitating ice in cloud free grid cells. The values shown in now Fig. 9 are filtered and scaled with the IFS cloud fraction, thus representing only the in-cloud IWC. This is consistent with ecRad's treatment of clouds. ecRad only calculates ice optical properties for grid cells which are cloudy (cloud fraction $> 0$) and have an IWC greater $10^{-9}\,\mathrm{kg\,kg^{-1}}$. All the IWC values in our case studies are above this value. Because of this and since there is no IWC threshold to differentiate between clear and cloudy skies, the sensitivity of the small value bins in the histogram cannot be explored.

*"4. l. 318-355 and Figure 9: The value of the IWC has an impact on the radiative flux and it is interesting to show the comparison between the measured values and those of the model. But the flux also depends on the vertical integral of the IWC (i.e. the ice water path, IWP). The IWP depends not only on the IWC but also on the way in which vertical overlaps occur. It would therefore be interesting to compare the IWP as well.*

This is a good suggestion, which lead us to include the IWP in Sect. 5.4 "IWC and $r_{\mathrm{eff}}$ input", where we switch the IFS IWC and $r_{\mathrm{eff}}$ for the VarCloud retrieved values. Therefore, we added another row to now Fig. 9 showing the IWP distribution for the two case studies and adjusted the figure caption accordingly. We also adjusted the introduction of Fig. 9 and added an explanation of what can be seen in the IWP distributions.

**Figure 9**: Added two new panels (c) and (d) showing the IWP distributions from the IFS and VarCloud for the case study areas in RF 17 and RF 18, respectively.

**L384-385**: Rephrased to *"Due to the temporal resolution of VarCloud, more points are available for the retrieval compared to the IFS while the IWP distributions naturally have less data points."*

**L389-L396**: Added *"For RF 17 only very low IWP values below $20\,g\,m^{-2}$ are observed in the VarCloud data, while the IFS also shows values up to $80\,g\,m^{-2}$. Nonetheless, most of the IFS values are also concentrated in the first two bins below $20\,g\,m^{-2}$ giving both distributions a strong positive skewness. This shows that, although the IFS predicts more smaller IWC values during RF 17, it overpredicts the IWP and thus the optical thickness as has already been shown with the solar transmissivity in Fig. 7. The histograms for RF 18 in Fig. 9 (d), in analogy to the IWC distributions, are flatter and show more large values. Apart from the VarCloud data showing more values in the lower bins between $20\,g\,m^{-2}$ and $40\,g\,m^{-2}$ and the IFS data having values above $90\,g\,m^{-2}$, which the VarCloud data is missing, the distributions are rather similar."*

*"5. l. 390-391: You talk about the effect of 'reice' on the model results, but what is the sensitivity of the estimated values to $r_{\mathrm{eff}}$? Are the values of $r_{\mathrm{eff}}$ used consistent with those of the model?"*

We assume, the first question addresses the similar analysis of the parameterization of $r_{\mathrm{eff}}$ as recommended by reviewer 1. We added a paragraph with a sensitivity study at the beginning of Sect. 5.2 concerning the $r_{\mathrm{eff}}$ parameterization. With this we could show that by varying $r_{\mathrm{eff}}$ in its allowed range, changes of the solar downward irradiance below cloud between $-5\,\%$ and $+35\,\%$ are possible.

**L347-L352**: *To explore the sensitivity of the ecRad simulations to $r_{\mathrm{eff}}$ the $12\,UTC$ time step from the below cloud section of RF 18 is taken and $r_{\mathrm{eff}}$ inside the cloud is varied in the possible range of values between $13\,\mu m$ and $100\,\mu m$. The simulations are preformed using the reference setup with the IFS IWC and the Fu-IFS ice optics parameterization. Comparing the solar downward irradiance below the cloud across all simulations shows a change between $-5\,\%$ to $+35\,\%$ with respect to the original value. Thus, we conclude that $r_{\mathrm{eff}}$ is indeed one of the driving factors in the ecRad simulations.*

We are not quite sure, what the second part of your question refers to. It could mean, whether the VarCloud retrieved $r_{\mathrm{eff}}$ values are within the range of the possible values predicted by the Sun (2001) parameterization. If that is the case, the question can be answered with a yes. The maximum value returned by Sun (2001) within the IFS is $100\,\mu m$ and all retrieved values are below that. The Sun (2001) parameterization could theoretically return larger values, but the implementation within the IFS caps the maximum value, because the ice optics parameterization by Fu (1996) would result in asymmetry parameters larger than 1 for $r_{\mathrm{eff}}$ larger than $100\,\mu m$.

Another interpretation of your question could concern the difference in definition of $r_{\mathrm{eff}}$ between the Sun and Rikus (1999) parameterization and the VarCloud retrieval by Ewald et al. (2021). Both $r_{\mathrm{eff}}$ definitions are either taken directly from Foot (1988) (VarCloud) or can be related to it (Sun and Rikus 1999). However, while Sun and Rikus (1999) assume hexagonal columns as a particle shape, Ewald et al. (2021) use the horizontally aligned oblate spheroid approximation from Hogan et al. (2012). This is due to the two papers trying to achieve quite different objectives. Ewald et al. (2021) want to retrieve microphysical properties from active remote sensing measurements and thus need to simulate radar and lidar signals, from which they then derive the IWC and $r_{\mathrm{eff}}$. Sun and Rikus (1999) on the other hand want to use the IWC and temperature predicted by a model as input to parameterize $r_{\mathrm{eff}}$. The parameterized $r_{\mathrm{eff}}$ can then be used in ice optics parameterization.

Hopefully, one of the two explanations answered your question.

**Revised figures**

[Figure]

Figure 1: Map of flight tracks with IFS predicted high cloud cover for 12 UTC, sea ice edge (80 % sea ice cover), mean sea level pressure isolines, dropsonde locations (red crosses), highlighted case study regions (orange), and LAGRANTO backward trajectories for (a) RF 17 and (b) RF 18. The box in panel (b) shows a zoom of the case study region with the above and below-cloud flight sections for RF 18. (c) and (d) False color corrected reflectance from MODIS on Terra using Band 3, 6 and 7 for RF 17 and RF 18, respectively, as provided by the Global Imagery Browse Services (GIBS) from NASA.

[Figure]

Figure 3: Measured downward and upward solar irradiance from BACARDI for the (a, b) above and (c, d) below-cloud sections of (a, c) RF 17 and (b, d) RF 18. Panels (e) and (f) show the solar transmissivity below cloud. The x-axis shows the distance traveled by HALO from the start to the end of the above-cloud section.

[Figure]

Figure 4: Atmospheric profiles of (a, c) air temperature and (b, d) relative humidity over ice from the IFS (grey lines) for the whole case study period (above and below-cloud section) along the flight track and the dropsondes (DS) deployed by HALO during the above-cloud section of (a, b) RF 17 and (c, d) RF 18. The black line indicates the flight altitude of HALO during the below-cloud section.

[Figure]

Figure 9: Probability density functions of (a, b) IWC with $1\,\mathrm{mg\,m^{-3}}$ binwidth, (c, d) IWP with $10\,\mathrm{g\,m^{-2}}$ binwidth and (e, f) $r_{\mathrm{eff}}$ with $4\,\mathrm{\mu m}$ binwidth for (a, c, e) RF 17 and (b, d, f) RF 18 of the IFS/parameterization output from the below-cloud section and the VarCloud retrieval. n depicts the number of points used in each histogram.

**References**

Baran, Anthony J., Peter Hill, David Walters, Steven C. Hardiman, Kalli Furtado, Paul R. Field, and James Manners. 2016. "The Impact of Two Coupled Cirrus Microphysics–radiation Parameterizations on the Temperature and Specific Humidity Biases in the Tropical Tropopause Layer in a Climate Model." *J Climate* 29 (14): 52995316. https://doi.org/10.1175/jcli-d-15-0821.1.

Ewald, Florian, Silke Groß, Martin Wirth, Julien Delanoë, Stuart Fox, and Bernhard Mayer. 2021. "Why We Need Radar, Lidar, and Solar Radiance Observations to Constrain Ice Cloud Microphysics." *Atmospheric Measurement Techniques* 14 (7): 5029–47. https://doi.org/10.5194/amt-14-5029-2021.

Foot, J. S. 1988. "Some Observations of the Optical Properties of Clouds. II: Cirrus." *Quart. J. Roy. Meteorol. Soc.* 114 (479): 145164. https://doi.org/10.1002/qj.49711447908.

Fu, Qiang. 1996. "An Accurate Parameterization of the Solar Radiative Properties of Cirrus Clouds for Climate Models." *J Climate* 9 (9): 20582082. https://doi.org/10.1175/1520-0442(1996)009%3C2058:aapots%3E2.0.co;2.

Hogan, Robin J., and Alessio Bozzo. 2018. "A Flexible and Efficient Radiation Scheme for the ECMWF Model." *J. Adv. Model. Earth Syst.* 10 (8): 19902008. https://doi.org/10.1029/2018ms001364.

Hogan, Robin J., Lin Tian, Philip R. A. Brown, Christopher D. Westbrook, Andrew J. Heymsfield, and Jon D. Eastment. 2012. "Radar Scattering from Ice Aggregates Using the Horizontally Aligned Oblate Spheroid Approximation." *Journal of Applied Meteorology and Climatology* 51 (3): 655–71. https://doi.org/10.1175/JAMC-D-11-074.1.

Sun, Zhian. 2001. "Reply to Comments by Greg m. McFarquhar on 'Parametrization of Effective Sizes of Cirrus-Cloud Particles and Its Verification Against Observations.' (October b, 1999,125, 3037–3055)." *Quart. J. Roy. Meteorol. Soc.* 127 (571): 267271. https://doi.org/10.1002/qj.49712757116.

Sun, Zhian, and Lawrie Rikus. 1999. "Parametrization of Effective Sizes of Cirrus-Cloud Particles and Its Verification Against Observations." *Quart. J. Roy. Meteorol. Soc.* 125 (560): 30373055. https://doi.org/10.1002/qj.49712556012.

Yi, Bingqi, Ping Yang, Bryan A. Baum, Tristan L'Ecuyer, Lazaros Oreopoulos, Eli J. Mlawer, Andrew J. Heymsfield, and Kuo-Nan Liou. 2013. "Influence of Ice Particle Surface Roughening on the Global Cloud Radiative Effect." *Journal of the Atmospheric Sciences* 70 (9): 2794–2807. https://doi.org/10.1175/JAS-D-13-020.1.